# Contrast and luminance adaptation alter neuronal coding and perception of stimulus orientation

Masoud Ghodrati [1,2], Elizabeth Zavitz [1,2], Marcello G.P. Rosa [1,2] & Nicholas S.C. Price [1,2]

Sensory systems face a barrage of stimulation that continually changes along multiple dimensions. These simultaneous changes create a formidable problem for the nervous system, as neurons must dynamically encode each stimulus dimension, despite changes in other dimensions. Here, we measured how neurons in visual cortex encode orientation following changes in luminance and contrast, which are critical for visual processing, but nuisance variables in the context of orientation coding. Using information theoretic analysis and population decoding approaches, we find that orientation discriminability is luminance and contrast dependent, changing over time due to firing rate adaptation. We also show that orientation discrimination in human observers changes during adaptation, in a manner consistent with the neuronal data. Our results suggest that adaptation does not maintain information rates per se, but instead acts to keep sensory systems operating within the limited dynamic range afforded by spiking activity, despite a wide range of possible inputs.

[1] Department of Physiology, Neuroscience Program, Biomedicine Discovery Institute, Monash University, Clayton, VIC 3800, Australia. [2] ARC Centre of Excellence for Integrative Brain Function, Monash University, Clayton, VIC 3800, Australia. Correspondence and requests for materials should be addressed to M.G. (email: masoud.ghodrati@monash.edu) or to N.S.C.P. (email: nicholas.price@monash.edu)

Our sensory systems receive a barrage of stimulation that continually changes along multiple dimensions and on multiple timescales. Even when looking around a simple scene, the receptive field of a single neuron in the visual system is stimulated by a dynamic sequence of spatial patterns, luminances, contrasts and colours. Considering just contrast and orientation, the two dimensions that profoundly affect the firing rates and response dynamics of neurons in the early visual system[1,2], the range of possible stimulus combinations vastly exceeds the limited dynamic range of any neuron's spiking output. One way in which individual neurons can better represent the current stimulus is to continuously update their limited response dynamics to account for the recent stimulus history. However, the mechanisms underlying this are unclear[3]. Further, it is unclear how changes along one stimulus dimension affect the neural coding properties of other dimensions, when individual neurons are continuously stimulated by a multidimensional feature space.

Adaptive mechanisms are prominent in all species and sensory modalities studied[3]. For example, throughout the visual system, luminance and contrast-gain control help to maintain perceptual sensitivity under different lighting environments by changing the temporal dynamics and gain of neuronal responses[4,5]. These mechanisms dynamically shift the operating point of neurons in a manner that maximises information transmission[6,7] or feature detection and processing[8,9].

Previous studies have focused on the effects of adaptation to a single stimulus dimension and how they affect the neural coding of that particular dimension. For example, we have examined how the exposure to a single motion direction affects the encoding of other motion directions, at the level of both single neurons and populations[10,11]. Others have examined how the exposure to changing distributions of stimulus statistics, such as stimulus speeds, luminances or sound intensities, affects the encoding of those specific stimulus dimensions[12–14]. In natural vision, neurons encode rich stimuli in a multidimensional feature space; yet it remains elusive how neurons dynamically encode each input dimension if stimuli are also changing on other dimensions. Put simply, how does the adaptation in one dimension affect the coding in another? Given the frequent variations in firing rates across the neuronal population due to changes in single dimensions, such as the mean luminance or contrast[15–17], an important question is how neurons can stably code information about the barrage of multidimensional sensory information in dynamic environments.

To address this question, we recorded the extracellular neuronal activity in the primary visual cortex (V1) of marmoset monkeys viewing a movie of sinusoidal gratings that changed the orientation every 16.7 ms, with concurrent changes in mean luminance or contrast every 5 s. This experimental design allowed us to investigate how the adaptation to one dimension (luminance or contrast) affects the neural coding of another dimension (orientation). Our study is the first to reveal how the orientation coding in V1 neurons is impacted by adaptation to presumably orthogonal stimulus dimensions. Although the encoding of luminance and contrast are critical functions of the visual system, here we are interested specifically in the encoding of orientation during adaptation; therefore, we treat luminance and contrast as nuisance variables in the statistical sense.

Using information-theoretic analysis and population-decoding approaches, we found that the ability of single neurons and neural populations to discriminate orientation is highly dependent on luminance and contrast. Our reverse-correlation analysis also showed that the temporal kernel of single-neuron orientation tuning changes during adaptation. More importantly, we found that orientation discriminability changes during adaptation periods that follow switches in luminance and contrast conditions in a manner

that closely reflects human perceptual thresholds. Interestingly, we found that the gain, but not the tuning bandwidth, of orientation-tuning curves for single neurons is multiplicatively scaled throughout the course of luminance and contrast adaptation. Our results suggest that the adaptation does not maintain information rates per se, but instead acts to keep sensory systems operating within the limited dynamic range afforded by spiking activity, despite a wide range of possible inputs.

## Results

**Luminance and contrast adaptation affect tuning not timing.** We recorded the extracellular neuronal activity in V1 of marmoset monkeys under sufentanil/$N_2O$ anaesthesia, in response to a movie of rapidly presented oriented gratings, the properties of which continually changed on rapid and slow timescales. We designed this switching paradigm to systematically study how adaptation to variations in one stimulus dimension affects the coding of other stimulus dimensions. Specifically, rapid variations in stimulus orientation occurred every 16.7 ms, while changes in mean luminance and contrast occurred every 5 s (Fig. 1a). This allowed us to examine how the neural information about orientation depended on steady-state luminance and contrast and on the time since specific switches in luminance and contrast have occurred.

In order to quantify the orientation tuning in a fine temporal detail, we used a reverse-correlation approach to estimate the probability at which each possible orientation occurred at all times preceding an action potential[18]. Applying orientation-reverse correlation in the context of adaptation to stimulus luminance and contrast allowed us to examine how adaptation impacts tuning over time. First, we compared the dynamics of orientation selectivity measured early (0–1.6 s) and late (3.4–5 s) after a luminance or contrast switch (Fig. 1a, b) and subsequently compared the dynamics of orientation selectivity between each of the four luminance–contrast conditions.

The strength of tuning, quantified as the maximum modulation in the linear kernel (Fig. 1c), changed over time and was significantly higher during the late phase of adaptation following a contrast increment, regardless of the luminance (Fig. 1d, $p < 10^{-7}$, signed-rank test). These changes in modulation during adaptation are consistent with the previous observations in the retina and lateral geniculate nucleus (LGN)[19,20]. Although the strength of orientation tuning changed with adaptation, we found no evidence for systematic changes in the width or the peak time of the temporal profile between the early and late phases after a contrast increment (Fig. 1e, f; see Supplementary Fig. 1 for other conditions). Furthermore, even when timing differences were significant, they were only of the order of 1–2 ms (Fig. 1e and Supplementary Fig. 1).

Comparing the orientation tuning across different luminance and contrast levels, rather than during adaptation to a fixed luminance and contrast, showed that maximum modulation, peak time and temporal width were consistently luminance and contrast dependent (Supplementary Fig. 2). Notably, low-luminance and high-contrast stimuli were associated with the largest modulation, shortest time to peak, and narrowest temporal widths, in agreement with previous reports[21].

These results show that the temporal properties of V1 neurons in response to stimulus orientation are strongly affected by luminance and contrast. Critically, while the peak time and temporal width appear to rapidly or instantly compensate for changes in luminance or contrast, the magnitude of the linear kernel changes more slowly during adaptation to a single condition. We asked how these properties affect the coding of orientation across time, given that luminance and contrast change on multiple timescales.

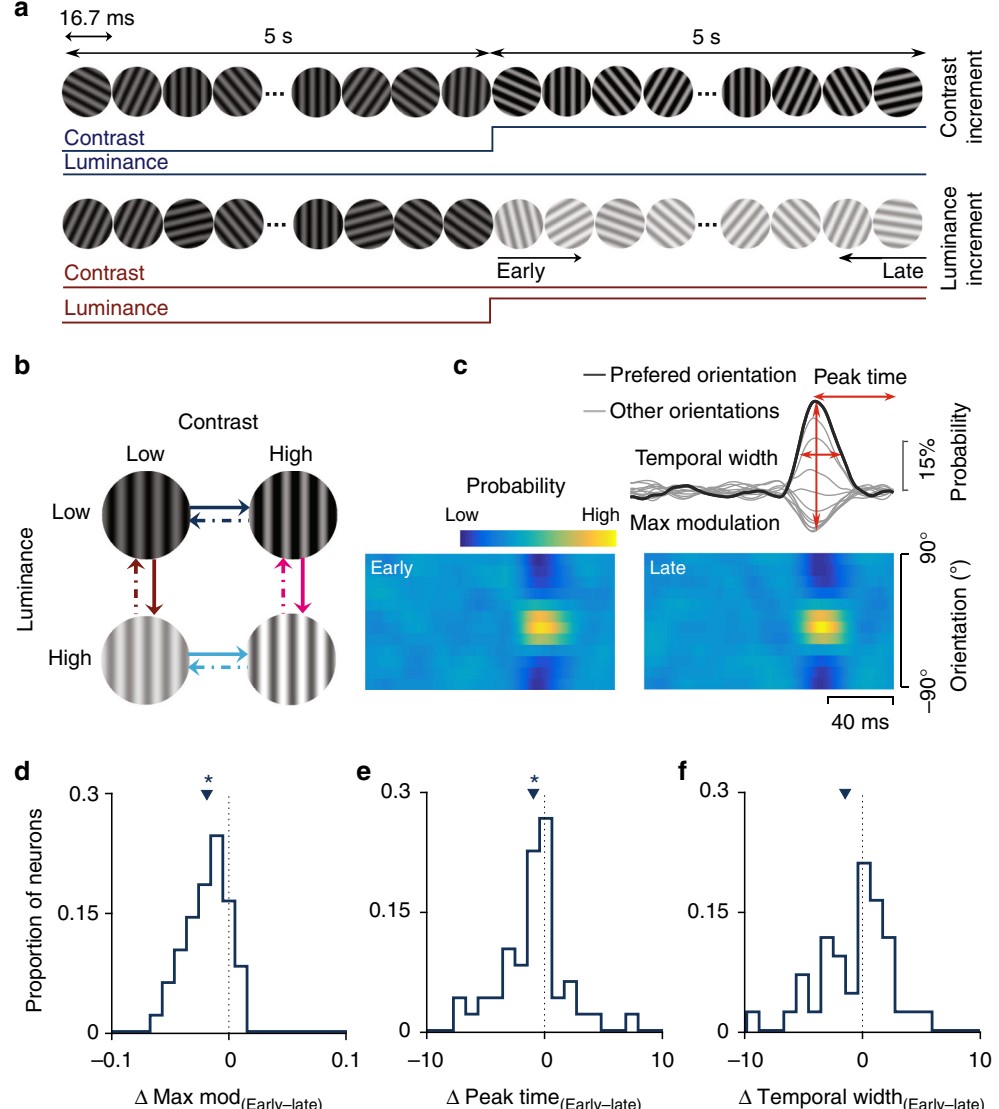

**Fig. 1** Stimulus design and linear-response characteristics of V1 neurons. **a** Switching stimulus paradigm showing two example conditions: a contrast increment and mean luminance increment. Every 16.7 ms (two frames at 120 Hz), a grating with randomly selected orientation, phase, and spatial frequency was presented. **b** The gratings' mean luminance or contrast was randomly selected from four conditions and was switched every 5 s, with a total presentation time of 60 min (maximum 720 switches). Here, we only analyse eight switches (indicated by coloured arrows), in which a single parameter changed. The colour code and line styles are consistent throughout the study. **c** The linear kernel quantifying the dynamics of orientation selectivity for a single neuron during the early (left, first 1.6 s) and late (right, last 1.6 s) phases of adaptation to high-contrast, low-luminance stimuli (i.e., the switch shown with solid dark blue arrow). Inset, temporal features were extracted from every linear kernel during adaptation. Population data of selective neurons comparing the differences in maximum modulation (**d**), peak time (**e**), and temporal width (**f**) between early and late phases for one adaptation condition (see the colour code). Small triangles indicate average; asterisks indicate significance ($p < 0.01$, $t$ test). $n = 390$

**Adaptation alters the coding efficiency of stimulus orientation.**
Changing the statistics of a specific stimulus dimension (such as luminance, intensity or speed) affects a neuron's coding efficiency of the same stimulus dimension[7,12,22]. However, it remains unclear if adaptation to one dimension in a multidimensional stimulus space affects the coding of other dimensions. To determine whether and how contrast and luminance adaptation affect the coding of orientation in V1 neurons, we calculated the mutual information (MI) between each neuron's spike count and stimulus orientation and examined the following: (1) how MI depended on luminance and contrast (Fig. 2a, b) and (2) how MI changed during the course of adaptation to a single luminance and contrast (Fig. 2c–f).

Initially, we examined MI during a late or steady-state time window, from 3.4 to 5 s after the stimulus switch. Information was higher in high-contrast conditions (Fig. 2a), regardless of the mean luminance (low luminance, $p < 10^{-30}$; high luminance, $p < 10^{-22}$; $t$ test). Surprisingly, information was also higher in low-luminance conditions (Fig. 2b; high contrast, $p < 10^{-31}$; low contrast, $p < 10^{-23}$; $t$ test). Although a high luminance is considered to be a stronger input to the retina and LGN than a low luminance, this result is consistent with the masking effect of high-luminance transients on neural responses and perception[17,23,24].

Given the significant information differences present between the different steady-state luminance and contrast conditions, we wondered how long it takes for these changes to manifest. To

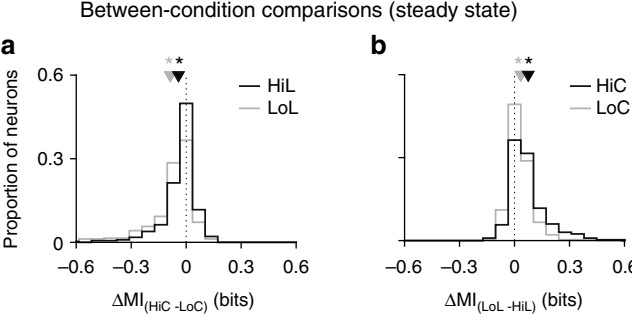

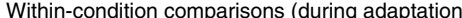

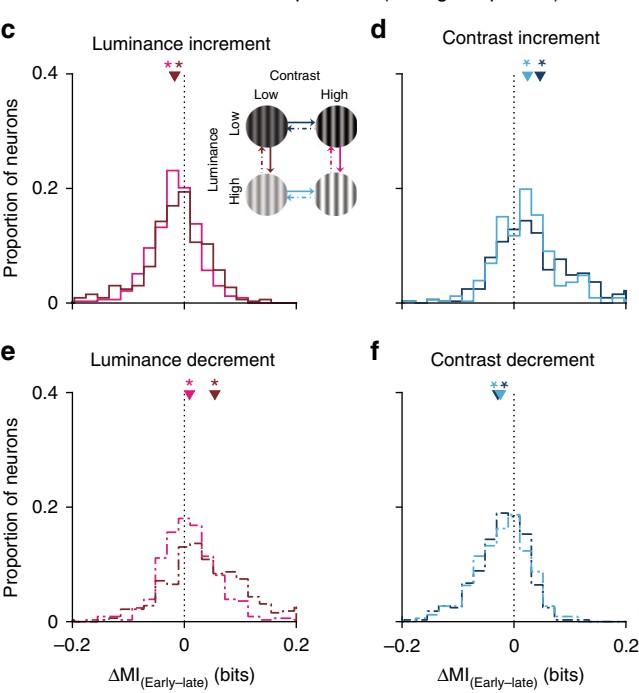

**Fig. 2** The difference in mutual information between different adaptation conditions. **a** Distribution of mutual information difference between high- and low-contrast conditions during the late or steady-state period (between-condition comparisons). Black: gratings had a high mean luminance (HiL); grey: gratings had a low mean luminance (LoL). **b** As in (**a**), but showing the distribution of Δ MI between high- and low-luminance conditions, for gratings with a high and low contrast. Black: gratings had a high contrast (HiC); grey: gratings had a low contrast (LoC). **c–f** Distribution of mutual information difference between early and late phases of adaptation for every luminance–contrast condition (within-condition comparisons). The solid lines indicate upward switches (e.g., low- to high-luminance switch in (**c**)); dashed lines indicate downward switches (e.g., high to low luminance in (**e**)). The inset in (**c**) demonstrates the colour code for luminance and contrast switches (e.g., dark blue indicates contrast increments when the gratings had a low mean luminance). Small triangles indicate the average mean of the distribution; asterisks indicate significance ($p < 0.01$, $t$ test). $n = 390$

characterise the effect of adaptation on the coding of stimulus orientation, we estimated the information conveyed by individual neurons about the stimulus orientation during early (0–1.6 s) and late (3.4–5 s) phases following a luminance or contrast switch. Following a contrast increment, regardless of the luminance, MI was significantly higher during the early than the late phase (Fig. 2d; low luminance, $p < 0.001$; high luminance, $p < 0.005$; signed-rank test). Similarly, MI was initially higher following a luminance decrement, for both contrasts (Fig. 2e; low contrast,

$p < 0.01$; high contrast, $p < 10^{-5}$; signed-rank test). However, the opposite trends were observed following contrast decrements (Fig. 2f; low luminance, $p < 10^{-8}$; high luminance, $p < 10^{-7}$; signed-rank test) and luminance increments (Fig. 2c; high contrast, $p < 10^{-8}$; low contrast, $p < 0.004$; signed-rank test). Note that we found the same qualitative trends when we used other time windows ranging from 0.2 to 2.4 s in duration to define our early and late periods. This suggests that luminance- and contrast-dependent changes in information about orientation are enacted on very short timescales.

**Firing-rate adaptation after luminance and contrast switches.** Given that the mean luminance and contrast affect firing rates, the above analysis motivated us to determine whether stimulus-induced changes in MI could be decoupled from changes in spiking rate. We averaged the spiking activity over dozens of repetitions of every unique luminance–contrast switch regardless of the changes across other stimulus dimensions (i.e., orientation, phase, and spatial frequency). This provided us with an estimate of the firing-rate variation that is only driven by switches in luminance and contrast. Effectively, we are averaging each neuron's orientation-selectivity profile at each time point relative to the luminance or contrast switch.

Luminance and contrast switches induced substantial changes in the firing rate of neurons, usually comprising an initial rapid increase or decrease in rate, followed by relaxation to an intermediate plateau following an exponential decay (Fig. 3 and Supplementary Fig. 3). The time constant of these decays depended on luminance and contrast; for example, the increase in firing rate following a contrast switch was significantly higher when gratings had a low luminance (Fig. 3b, dark-blue trace) rather than high luminance (Fig. 3b, light-blue trace; $p < 0.01$, signed-rank test). Time constants were also significantly longer for contrast decrements than increments (compare Fig. 3b with Fig. 3d; $p < 0.01$, signed-rank test; also see Supplementary Fig. 3). This asymmetry in the time course of firing-rate changes is consistent with the adaptation to variations in the statistics of auditory[12], visual[19], and tactile[22] stimuli in sensory areas. It is also apparent that a higher mean luminance, during both contrast increments and decrements, evoked weaker activity than a low luminance (compare dark-blue and light-blue traces in Fig. 3b, d).

Following a mean luminance increment, firing rates displayed a transient reduction followed by an exponential recovery lasting several seconds (Fig. 3a, Supplementary Fig. 3). Unusually, when the gratings had a low contrast (Fig. 3a, pink trace), there was a biphasic response in 30% of neurons, with a small and rapid change in mean firing rate immediately after a luminance switch. This biphasic response is similar to the impulse response of the visual neurons to a luminance switch[14,17].

Surprisingly, the firing rate rapidly increased following a switch from high to low mean luminance, regardless of the contrast (Fig. 3c). The time constant of increase and the subsequent exponential decay were in a similar range to contrast increment (Fig. 3b and Supplementary Fig. 3). We also observed a similar asymmetry in firing-rate adaptation during luminance increments and decrements to that following contrast switches (Fig. 3b), consistent with previously reported data[7,12,14,19,22].

**Luminance and contrast switches reduce neural variability.** Neural responses to multiple repetitions of the same stimulus are variable, and the neural coding efficacy depends on this variability. While recent studies have demonstrated that response variability that is correlated between neurons can surprisingly enhance neuronal coding[11,25], a higher variability in the responses of individual neurons, often quantified using the Fano

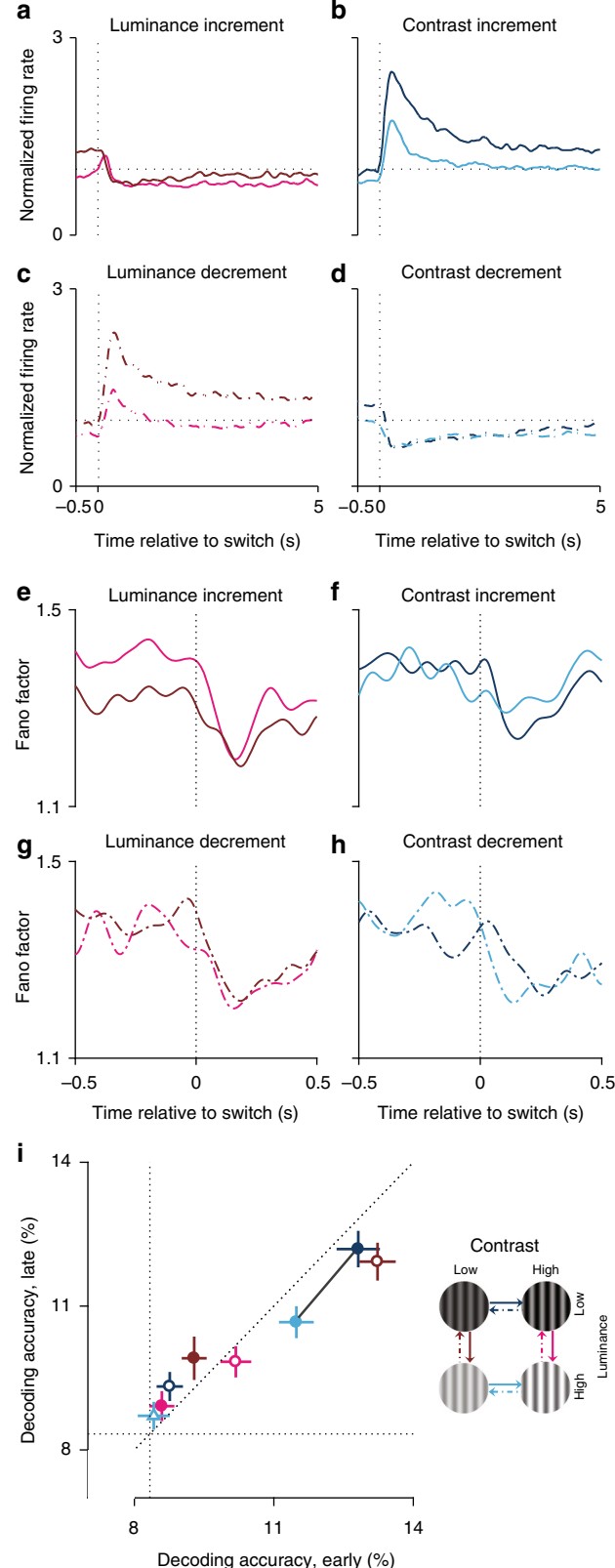

**Fig. 3** Firing-rate adaptation, neural response variability, and population coding. The average normalised population firing rate during luminance-increment (**a**), luminance-decrement (**c**), contrast-increment (**b**), and contrast-decrement (**d**) switches. Each PSTH was normalised relative to a shuffled PSTH, generated by shuffling the spike times. The shuffling process was repeated 50 times while the spike times across the entire 60-min recorded data were randomised in each run. For visualisation, PSTHs have been convolved with a Gaussian window (50 ms). **e–h** As in (**a–d**), but for Fano factor. **i** Decoding accuracies early and late following each of the eight luminance–contrast switches. Each data point shows the average and standard deviation of decoding accuracy for one switching condition (averaged over 15 runs). Solid points indicate upward switches; empty points indicate downward switches. The black line connects data points associated with low vs high luminance, when the contrast is high. The inset demonstrates the colour code for each luminance and contrast switch. Decoding algorithm: linear discriminant analysis (LDA); the number of neurons = 50 randomly selected out of 390 neurons; width of spike-counting window = 15 ms; and the number of random runs = 15. Error bars are standard deviations. Support vector machine (SVM) classifier provided very similar results. PSTH, peri-stimulus time histogram

underlying rate changes throughout our measurement window), we observed a consistent pattern of FF variations across different conditions (Figs. 3e–h). Overall, FF decreased immediately after any switch in luminance and contrast, consistent with the previous reports that changes in stimulation conditions quench neural variability[26]. The FF decreased and then rapidly recovered in all cases after a luminance or contrast change, whereas the firing rate changed more slowly and could either decrease or increase (Fig. 3 and Supplementary Fig. 3). Given these observations, if population decoding is primarily affected by trial-to-trial variability, it should always improve immediately after a change in the stimulus.

**Coding of stimulus orientation by neural populations**. We found that the information conveyed by individual neurons about stimulus orientation is strongly affected during adaptation. It is, however, unclear how the coding of orientation by a neural population is affected by changes over time in firing rates and trial-to-trial variability at the level of individual neurons, and whether the variability between neurons can be overcome at the population level. Moreover, neurons vary in their preferences, temporal dynamics, and adaptation properties. Therefore, we used a population-decoding approach to quantify orientation discriminability, asking how the decoding accuracy (1) is affected by different luminance and contrast conditions and (2) changes during the course of adaptation to a single luminance and contrast.

The results revealed a clear difference in discriminability between the luminance–contrast conditions and during the course of adaptation. Overall, higher contrasts led to a better orientation discriminability while higher luminance levels led to a poorer discriminability (Fig. 3i; e.g., compare dark- and light-blue data points, connected with a black line, for low vs high luminance). Moreover, the adaptation following a contrast increment decreased the orientation discriminability of neural populations while the adaptation following a contrast decrement led to higher discriminability (Fig. 3i; $p < 0.0001$, signed-rank test; e.g., filled blue data points associated with contrast increments fall below the line of unity). This was the opposite for luminance switches, as adaptation following luminance increments and decrements increased and decreased the discriminability, respectively (Fig. 3i; $p < 0.0001$, signed-rank test; e.g., filled brown and

factor (FF), can only impair encoding. We thus calculated the FF of neural responses within a sliding 50 ms time window from 500 ms before to 500 ms after luminance or contrast switches.

As with our previous analysis of firing rate, this approach ignores changes in the other dimensions of the stimulus. Despite the noisy measurement of FF using this method (because the

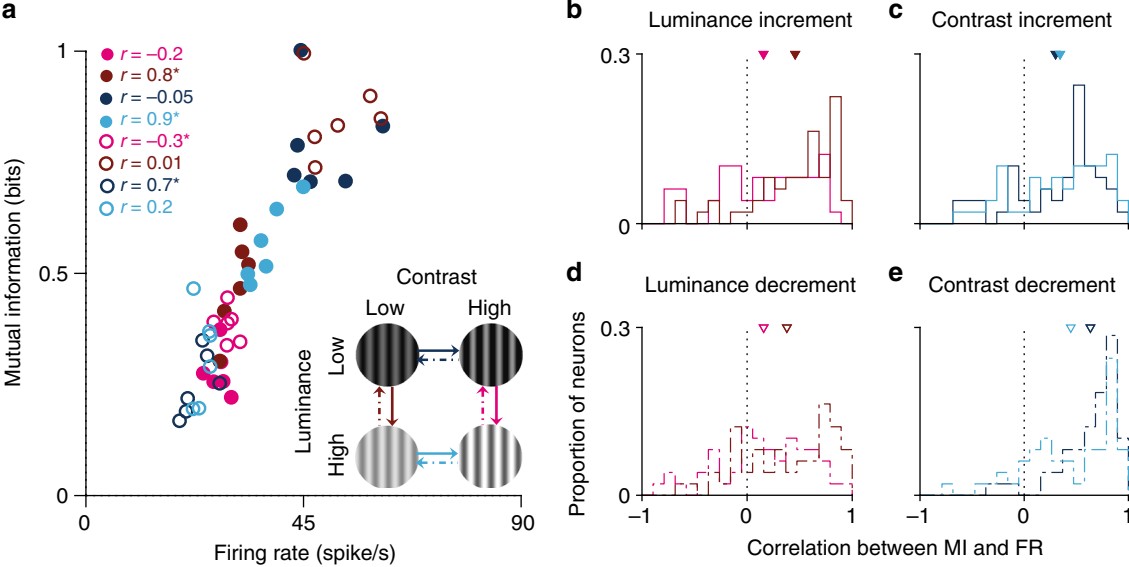

**Fig. 4** Changes over time in mutual information (MI) and firing rate (FR). **a** MI and firing rate were measured in six consecutive time windows during adaptation to each luminance–contrast condition. Each data point shows the result in a single time window, colour coded according to the stimulus condition. Correlations were calculated between the six data points for each stimulus condition. Asterisks indicate significance ($p < 0.05$). Solid circles indicate upward switches in luminance or contrast; empty circles show downward switches. **b–e** Population summary of correlations for each switch type. $n = 390$

pink data points associated with luminance increments fall above the line of unity). Similar decoding results were evident when we changed many parameters of the decoders, including different time windows for defining the early and late phases, the number of neurons in the decoder, the type of decoder, and the width of the readout, or spike-counting, window (Supplementary Figs. 4 and 5).

To compare the size of the changes in orientation coding at the single-neuron and population levels, we calculated the MI ratio (early relative to late) for each stimulus switch and compared it to the corresponding ratio of decoding performance (early relative to late). Across the eightswitches, these ratios were strongly correlated ($r = 0.87$, $p = 0.004$), with a slope of 0.4. As much as these ratios can be compared, this suggests that the adaptive changes over time are of similar magnitudes at the single-neuron and population levels. Further, coding of stimulus orientation by neural populations is substantially affected by adaptation to luminance and contrast, and this adaptive coding tracks the direction of changes in spiking rate, not response variability.

**Coding of stimulus orientation by individual neurons.** Previous studies of adaptation to stimulus statistics have shown that even though firing rates markedly change in association with switches in stimulus variance, the information rate of individual neurons (measured as bits per spike) is almost unaffected[7,22]. Our above analysis demonstrated that decoding of orientation by the neuronal population is substantially affected by luminance and contrast adaptation. To clarify this apparent conflict, we estimated the information conveyed by individual neurons during adaptation with a finer temporal resolution. Figure 4a illustrates, for a sample neuron, that the MI between stimulus orientation and spiking activity is also affected by adaptation. Here, we calculated the MI in six equal time windows that spanned the 5-s adaptation period following each of the eight stimulus switches (see also Supplementary Fig. 4). There are large, significant correlations between firing rate and MI for both individual stimulus conditions and when all the stimulus conditions are considered collectively. These correlations are also evident across the

population of neurons, and the average correlations are significantly greater than zero in all cases (Fig. 4b–e, $p < 0.01$, $t$ test). MI also changed during adaptation when we normalised information by spike count (bits per spike) rather than simply considering information (bits).

**Luminance and contrast switches modulate perception.** The changes in decoding performance and MI observed after switches in luminance and contrast suggest that orientation discrimination thresholds should be improved following contrast increments and luminance decrements. To assess this, we conducted human psychophysical experiments in which observers reported the relative orientation of two gratings, each presented for 200 ms, separated in time by a noise mask (Fig. 5a). On each trial, discrimination judgments were performed either early (0.2–1.2 s) or late (5–6 s) after a switch in luminance or contrast.

Orientation discrimination was enhanced immediately after a contrast increment, reflected in significantly higher discrimination thresholds during the late vs early phase (Fig. 5b, for a single subject, $p < 0.001$, bootstrap test and Fig. 5f, for all subjects, $p < 0.005$, signed-rank test, $n = 14$ observers). For 12 out of 14 observers, these increases in discrimination thresholds were individually significant ($p < 0.001$, bootstrap test), meaning that the majority of subjects performed significantly better in orientation discrimination during the early, compared to the late, phase. Similarly, discrimination accuracy was higher immediately after a luminance decrement, with discrimination thresholds significantly higher in the late vs early phase (Fig. 5e, $p < 0.01$, signed-rank test, $n = 7$ observers). In this experiment, all observers individually showed a significantly better discriminability in the early phase than in the late phase (Fig. 5e, $p < 0.001$, bootstrap test). On the other hand, we found no systematic changes in discrimination thresholds during the late vs early phase of luminance increments (Fig. 5c, for a single subject and Fig. 5d, for all subjects, $p > 0.05$, signed-rank test, $n = 7$ observers) and contrast decrements (Fig. 5g, $p > 0.05$, signed-rank test, $n = 8$ observers).

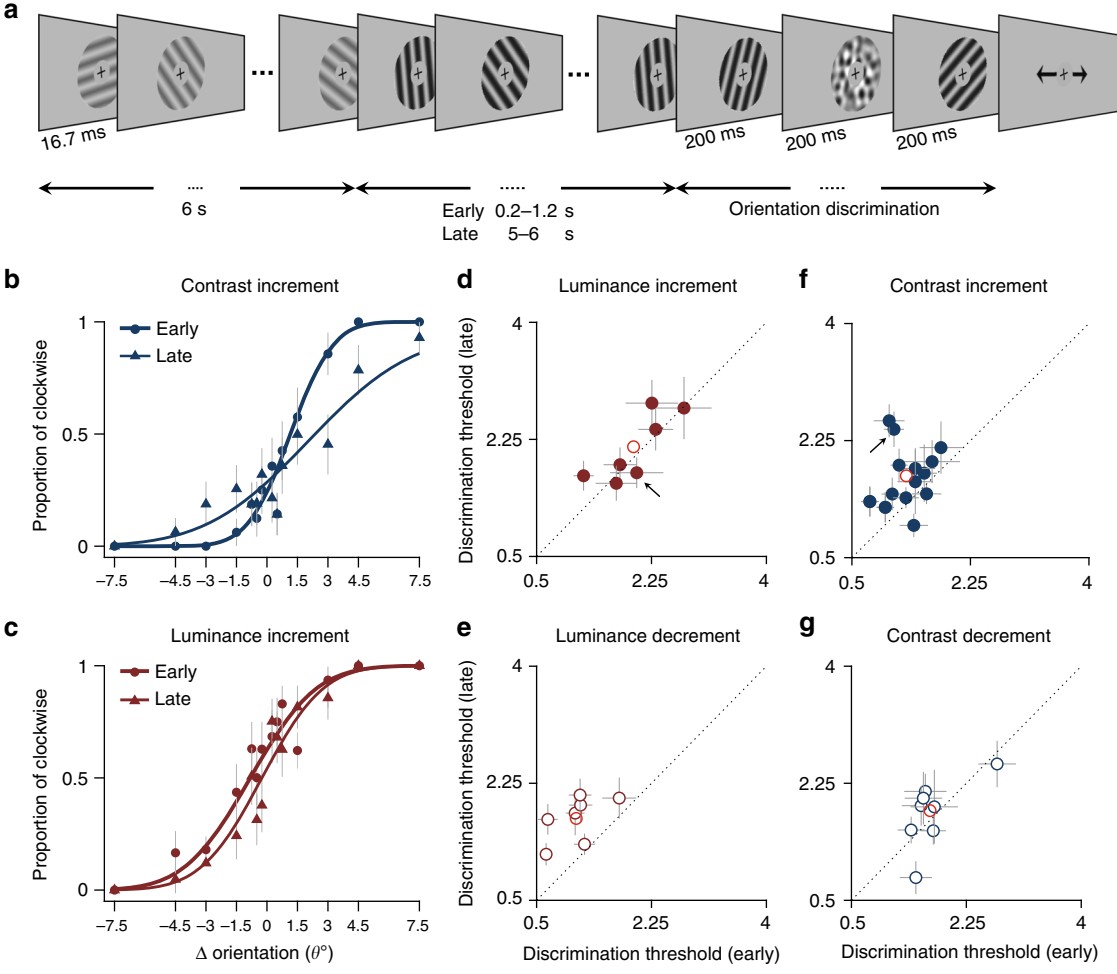

**Fig. 5** Psychophysical measurement of orientation discrimination. **a** The experimental paradigm. Each trial consisted of a short movie of rapidly presented sinusoidal gratings with random orientations and phases (each for two monitor frames, 16.7 ms). Subjects fixated on the cross throughout the trial. Subjects performed an orientation discrimination task after which the contrast (luminance) of the gratings changed to a higher contrast (lower luminance). The orientation discrimination was performed either 0.2–1.2 s (early trials) or 5–6 s (late trials) after the switch. **b**, **c** Psychometric data with cumulative Gaussian fit for a single subject performing the contrast- (**b**) and luminance-increment (**c**) tasks. **d–g** Comparison of discrimination thresholds for early and late phases of four different luminance- and contrast-adaptation conditions (the number of participants from (**d–g**): n = 7, 7, 14, and 8). The small arrow in (**d**) and (**f**) refers to the participant on the left, and the red circle shows the average threshold. Error bars show bootstrap standard deviation

These results are broadly consistent with our physiological data, in which the neural population-decoding accuracy was significantly higher during the early vs the late phase following a contrast increment and luminance decrement (Fig. 3). Despite this, an important methodological difference between the physiological and psychophysical studies limits the straightforward comparison of their results. In our physiological study, we used phenylephrine and atropine eye drops to dilate the pupils. This inactivates the pupillary light reflex, which modulates the amount of light reaching the retina.

To determine if differences in discrimination thresholds are related to pupil-size modulations, we monitored the pupil size in 15 observers (Supplementary Fig. 6). The pupil size was briefly, and only slightly, changed following contrast switches, but showed immediate and robust dilation or constriction following luminance switches. Contrast increments are therefore associated with consistent changes in perceptual and neuronal discrimination performances, in the absence of associated pupillary changes. However, while luminance decrements were also associated with consistent changes in perceptual and neuronal discrimination, they were only accompanied by pupillary dilation in the human observers. This means that although the observed changes in

neural coding can only arise as a result of cascading neural processes in the visual hierarchy (because the pupils were permanently dilated in the marmosets), the changes in human perceptual performance could reflect these same neural processes or simply the effects of pupillary dilation.

**Additive and multiplicative modulation of orientation tuning.** What causes the neurons to change their coding ability during adaptation? Coding properties of neurons can be affected by factors such as trial-to-trial variability and multiplicative and additive modulations in neural response populations. We earlier showed that trial-to-trial variability deceased immediately after luminance and contrast switches for all conditions (Fig. 3e–h), but the decoding accuracy did not always follow the same trend (Fig. 3i, Supplementary Figs. 4 and 5). Here, we asked whether other factors such as multiplicative and additive gain modulations could lead to the differential stimulus coding of orientation during adaptation. Note that multiplicative modulations change the response amplitude of single neurons without affecting their tuning selectivity[15,27]. However, such modulations of tuning functions become important when decoding the population

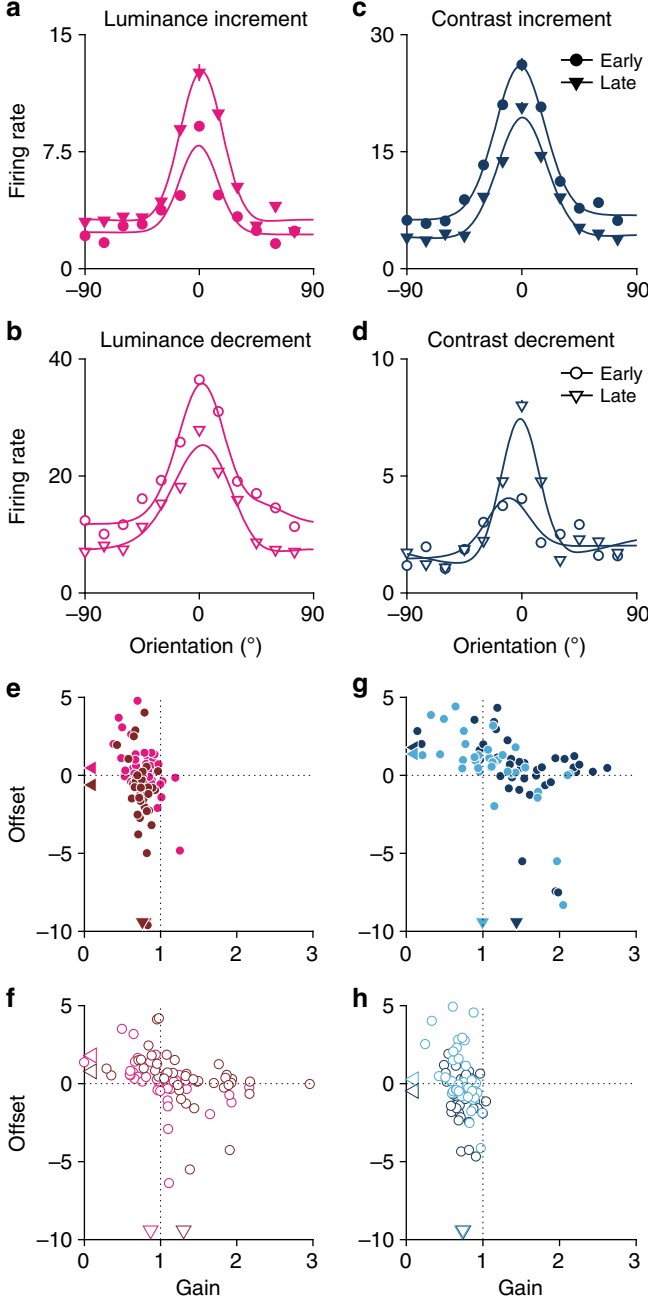

calculated multiplicative and additive modulations for the 50 neurons that were highly selective for all stimulus conditions. Working with this reduced dataset is necessary to allow direct within-condition comparisons. In this population, most neurons had strong gain changes during adaptation while the additive effect was minimal (Fig. 6e–h and Supplementary Fig. 7). For example, after adaptation to a luminance increment, regardless of the contrast, the gain of almost all neurons decreased, with no significant change in the additive modulation or offset (Fig. 6e, high contrast, $p < 10^{-17}$; low contrast, $p < 10^{-8}$, $t$ test). Similar changes were evident after a contrast decrement, regardless of luminance (Fig. 6h, high luminance, $p < 10^{-11}$; low luminance, $p < 10^{-14}$, $t$ test). Note that gain modulation is the gain change between the early and late phases of adaptation.

We observed small gain enhancements during adaptation to contrast increments (Fig. 6g; low luminance, $p < 0.05$; high luminance, $p > 0.05$, $t$ test) and luminance decrements (Fig. 6f; high contrast, $p < 0.05$; low contrast, $p > 0.05$, $t$ test). We did not find any significant additive modulation across the population during adaptation to these conditions (except in high-contrast, low-luminance conditions; Fig. 6f, $p < 0.01$, $t$ test). To further examine these effects at the level of individual neurons, we compared four models for predicting the tuning curves of each neuron at early and late phases. Each model comprised two von Mises functions (one for the early and the other for the late phase of the response) that were either identical, independent, matched in gain, but with independent offsets, or matched in offsets but with independent gain. For the cells that showed significant modulation in orientation tuning between the two response phases (i.e. model 1 was the poorest fit), we found that 17% were best explained with a combination of additive and gain modulations, 63% were best explained with only gain modulation, and 20% were best explained with only additive modulations. This suggests that it is primarily gain modulation, and to a lesser extent additive modulation, that explains the difference in orientation discriminability and information rate during adaptation.

We also calculated the gain and offset modulation with a finer temporal resolution during adaptation. To this end, we calculated the changes in the population-tuning curve across six consecutive non-overlapping time windows during adaptation following each type of switch (Supplementary Fig. 7). Our analysis showed that the gain and offset systematically change during adaptation (Supplementary Fig. 7). In particular, we found that the modulations during adaptation are not purely multiplicative or adaptive; however, the multiplicative modulations are more pronounced across single neurons.

## Discussion

In this study, we asked how principles that have been developed from studies of adaptation to a single stimulus dimension can be used as the foundation to track neural response selectivity under a more complex stimulation paradigm. In particular, how does adaptation to one stimulus dimension affect the coding of another dimension? To address this question, we investigated how the adaptation to mean luminance and contrast reshapes the encoding properties of V1 neurons and how this change in encoding properties alters the information about stimulus orientation that is available to downstream decoding neurons. We designed a switching stimulus paradigm in which stimulus dimensions varied on fast and slow timescales, a situation that happens in our everyday life as we step from bright sunlight into an indoor office, or simply change our point of gaze from glaring sunshine to the adjacent shade. We showed that adaptation to slowly varying stimulus features can alter the coding properties of

**Fig. 6** Gain and offset modulations of orientation tuning. **a–d** Tuning functions of a sample neuron during early and late phases of adaptation. Each panel shows a single switching condition. Solid lines show the best-fit von Mises function. **e–h** Population summary of gain and offset modulations between early and late phases for the eight switch types. Triangle markers indicate the mean. In all plots, only highly orientation-selective neurons are shown ($n = 50$)

activity, in part, because scaling up neural spiking rates disproportionately scales up the trial-to-trial variability, reflecting non-Poisson statistics[28].

We found that neuronal tuning functions varied greatly between early and late phases of adaptation. The sample neuron depicted in Fig. 6a–d highlights the range of effects commonly observed during different adaptation conditions. For example, the gain substantially increased after adaptation to a high luminance (Fig. 6a), while adaptation to a low luminance led to a compound effect of multiplicative and additive modulations (Fig. 6b). We

individual neurons and neural populations for a different rapidly varying stimulus dimension. In particular, we showed that following changes in luminance and contrast, the ability of neurons to code stimulus orientation dynamically changes, and these changes predict a novel form of perceptual adaptation in human orientation-discrimination thresholds. The dynamic coding in neural responses can be explained by the adaptive gain rescaling of individual neurons in the population.

Rapid adaptive stimulus-dependent changes in the filtering properties of neurons have been demonstrated in the visual[9,19,20], auditory[12,29] and somatosensory pathways[3,22]. Adaptation to changes in stimulus luminance and contrast also affects both linear and nonlinear filtering stages of a cascade model of retina[7,30,31] and LGN[20,32]. For example, adaptation to low-contrast stimuli can lead to faster temporal dynamics of the linear filter and a lower gain of the nonlinear filter. These studies examined how changing the statistics of one dimension affected the encoding of the same dimension. Our stimulus design provided us with the advantage that we could study the time frame over which such adaptive changes occur, as we had enough measurement time and stimulus repetitions to be able to use reverse-correlation analysis to robustly capture the aspects of dynamic feature selectivity of neurons.

Our analysis showed that the temporal dynamics of neurons' linear responses, measured as peak time and temporal width, was determined by mean luminance and contrast, but these changes manifested extremely rapidly and remained largely unaffected during the course of adaptation to a constant luminance or contrast. This happened despite the fact that the firing rate of neurons substantially varied during adaptation. Further analysis of linear-response kernels, however, showed significant changes in the amplitude of kernels (maximum modulation) during different phases of adaptation to luminance and contrast. For example, our results revealed that an increment in contrast can trigger a significant change in the shape of the linear kernel during the early phase of adaptation compared to the late phase, suggesting a gradual increase in the amplitude of the kernel during adaptation to a high contrast. However, following a contrast decrement, the kernel amplitude did not significantly change, likely because of the fact that changes in response amplitude after a contrast decrement are small and have a very slow timecourse of recovery. Overall, larger kernel amplitudes indicate that, given the occurrence of a spike, the stimulus orientation is known with a higher probability. Moreover, the amplitude modulation of the linear kernel was largely dependent on adaptation condition. The observed changes in the kernel amplitude during adaptation to contrast decrement and increment are consistent with those in the other studies in the retina[19] and auditory cortex[29]. However, changes in other temporal aspects of the linear filter, such as peak time and temporal width, have also been reported in auditory[29] and early visual pathways[20], which we did not observe in our study. We think that the difference in stimulus type might underlie this discrepancy, as these other studies have used white noise or natural stimuli, which are very different to the movie of rapidly changing gratings that we presented. For example, there are strong spatial and temporal correlations in the neighbouring pixels in natural movies. In summary, our results suggest that neurons do not employ a simple static-coding strategy but dynamically change their receptive-field profile to adjust to constant and rapid changes in the statistics of input stimuli. This adds to the previous studies showing that neural responses are poorly predicted using a static receptive field[9].

Generally, a switch in stimulus variance or mean leads to a transient change in the firing rate of neurons followed by an exponential[14,33] or power-law decay to steady state[7,34]. The time constants of firing-rate adaptation in our study are in a similar range to those in the previous studies of contrast and luminance adaptation in the visual pathway. These time constants are dependent on factors such as the period of the stimulus-switching paradigm[7,34] (but see ref. [12]) and the noise level of the input signal[14], suggesting that neural responses are dependent on the history of stimulus variations. In our study, we did not explore the effect of different switching periods and noise level on the time course of adaptation and feature selectivity, but we observed that neurons did not have a single time constant in all adaptation conditions, suggesting that recent changes in stimulus statistics affect neural response dynamics in V1.

Firing-rate adaptation does not follow a symmetric dynamics during upward and downward changes in stimulus statistics. Such an asymmetry has been observed in many sensory areas, suggesting this as a general property for adaptation[7,12,14,19,29,33–35]. We observed a similar asymmetric dynamics in adaptation to contrast increment and decrement, with a significantly longer time constant of adaptation during the switch to high than low contrasts. The trend was, however, different during the mean luminance switch, in which adaptation to a low mean luminance was faster than high luminance, suggesting that it takes longer for the neurons to recover after a switch to a high mean-luminance stimulus. This long-lasting suppression does not agree with the previous studies in the retina. One possible reason for this discrepancy may be the differences in response and anatomical properties between retina and V1 as a dense network of different excitatory and inhibitory connections. As one study has shown, while the responses of LGN neurons are elevated after a luminance transient, the responses of V1 neurons are significantly suppressed, and the selectivity of neurons to ongoing stimulus orientation is delayed[17]. Such a mechanism has been attributed to changes in cortical inhibition[17]. In this context, the responses of our recorded neurons to luminance switch are consistent with the literature. At the perceptual level, a sudden change in luminance also supresses the detectability and discriminability of visual targets in human observers[24].

Previous studies of adaptation using switching-stimulus paradigms have shown that the information content of single neurons (bits per spike) about the contrast (variance) and luminance (mean) of an input signal is largely unaffected around the time of a switch in variance[7,22]. Here, we have shown that the MI between spiking activity and stimulus orientation is higher when luminance is low (or contrast is high), consistent with the masking effect of luminance increments reported in electrophysiological[17,23,36] and perceptual studies[37]. The aim of our study was to explore the effect of adaptation to luminance and contrast on the coding of stimulus orientation. From the perspective of encoding stimulus orientation, changes in luminance and contrast are problematic, making them nuisance variables. Despite this, neurons in most visual areas actually carry information about luminance and contrast, and this information can be reliably decoded. While not the focus of our study, this is evident in our data as differences in the steady-state firing rate after adaptation to luminance and contrast (e.g. the mean firing rates for low and high contrasts are different and thus decodable).

Calculating the information content of individual neurons during the course of adaptation, however, showed that MI was significantly affected during adaptation to a single-luminance and -contrast condition. For example, the information conveyed by neurons about stimulus orientation was higher during the early than the late phase of an upward switch in contrast, suggesting that adaptation to contrast increments reduced the amount of information. Conversely, information about orientation increased

during adaptation to a contrast decrement. Although these changes in information might seem contradictory to the findings of the studies mentioned above, our result is comparable to some results reported from V1 and middle temporal area (MT) regarding the large contribution of onset transient to sensory discrimination[38,39]. Here, a switch to a new luminance–contrast can be considered as the onset of a new stimulation regime.

While single neuron-level mechanisms can account for some adaptive properties described in our study, cortical neurons operate in a highly interconnected neural circuitry. Therefore, the adaptive properties of a single neuron can induce substantial changes in information processing when interpreted in the context of neural population activity[10,11,40]. Moreover, our information-theoretic analysis mostly relies on orientation-selective neurons while non-selective neurons can affect the population code[11,41] and, likely, perception. A simple way to address this issue is using population decoding to investigate if the observed effects of adaptation at the single-neuron level are also evident at the population, or circuit, level. Our analysis showed that the decoding accuracy is significantly luminance and contrast dependent and strongly affected by adaptation. Adaptation to a high contrast (following a contrast increment) and a low luminance (following a luminance decrement) decreases the discriminability of stimulus orientation across neural populations, while adaptation to a high luminance and a low contrast increases the discriminability. These different changes in discriminability after adaptation do not perfectly agree with studies suggesting that adaptation always improves discriminability[42,43], as we found that discriminability can largely be stimulus or task dependent (e.g., compare the decoding accuracy between upward and downward contrast switches).

We also found changes in discrimination thresholds between the early and late phases in our psychophysical measurements for two switching conditions, suggesting that the significant modulations in firing rate and neural discriminability after the switch have a perceptual correlate in human subjects. While we cannot rule out the role of pupil-size variations in the perceptual luminance-adaptation effects we observed, two factors make it likely that the neural adaptation observed following both contrast increments and luminance decrements can account for the changes in human psychophysical performance. First, we observed a significant difference in perceptual thresholds between the early and late phases following contrast increments, even though the pupil size remained mostly unchanged. Second, following a luminance increment, pupil size changed substantially over time, but we observed a relatively little change in discrimination thresholds.

Collectively, we have shown that orientation discriminability is luminance and contrast dependent, changing over time due to firing-rate adaptation. This is accompanied by changes in the information available about orientation from neural spiking, attributable to adaptive gain modulation. Multiple cellular and network mechanisms may account for an adaptive gain control in the cortex. At the level of single neurons, adaptation largely relies on cellular mechanisms (e.g., $Na^+$-activated and $Ca^{2+}$-activated $K^+$ currents)[44] and synaptic depression[45]. Adaptive mechanisms can also be derived from network dynamics and recurrent inhibition, which can produce neuronal response dynamics that vary over a range of timescales[3,46]. In a highly interconnected network such as V1, lateral inhibition or modulatory feedback can also account for adaptive gain control[47–50]. Future studies should identify the distinct contributions of inhibitory and excitatory neurons to the changing feature selectivity that occurs during adaptation in order to link single-neuron and circuit-level mechanisms to perceptual outcomes.

## Methods

**Surgery and animal preparation**. We recorded the extracellular activity from V1 neurons in three anaesthetised male marmoset monkeys (*Callithrix jacchus*). Experiments were conducted in accordance with the Australian Code of Practice for the Care and Use of Animals for Scientific Purposes, and all procedures were approved by the Monash University Animal Ethics Experimentation Committee. The details of animal preparation, surgical procedure, and electrophysiology in marmoset monkeys followed our previously published protocols[51]. Animals were premedicated with atropine (0.33 mg kg$^{-1}$) and diazepam (0.4 mg kg$^{-1}$) and anaesthesia was subsequently induced with alfaxalone (Alfaxan, 8 mg kg$^{-1}$), after a tracheotomy, vein cannulation and craniotomy to be performed. After the completion of all surgical procedures, the animal was administered an intravenous infusion of pancuronium bromide (0.1 mg kg$^{-1}$ h$^{-1}$) combined with sufentanil (6–8 µg kg$^{-1}$ h$^{-1}$) and dexamethasone (0.4 mg kg$^{-1}$ h$^{-1}$), and was artificially ventilated with a gaseous mixture of nitrous oxide and oxygen (7:3). Pulse oxygenation, heart rate, body temperature and the level of cortical spontaneous activity were continuously monitored. The administration of atropine (1%) and phenylephrine hydrochloride (10%) eye drops resulted in mydriasis and cycloplegia. Protection of the corneas from desiccation and focusing on the stimulus monitor were achieved using hard contact lenses selected by retinoscopy.

**Electrophysiology, data acquisition, and pre-processing**. Most of the recordings were performed using single-shank linear multielectrode arrays (A1x32; Neuro-Nexus, Ann Arbor, MI, USA). Contacts on the array surface were collinear with 50 µm spacing, spanning all cortical layers. The data from one animal (monkey 2) were recorded using an Utah array, which consists of 96 electrodes arranged in a $10 \times 10$ grid, with each electrode separated by 400 µm (Blackrock Microsystems). Electrophysiological data were recorded using a Cerebus or Cereplex system (Blackrock Microsystems, MD) with a sampling rate of 30 kHz. The recordings were obtained from the region of V1 representing the central 10° of the visual field, in the exposed surface of the occipital operculum.

To detect single neurons and multi-units, we performed offline spike detection and sorting separately for each channel. Potential spikes were first identified based on threshold crossings, which were manually set during recording. Each spike waveform was normalised by its energy, and then principal component analysis was performed on all spike waveforms recorded from a channel. Normalisation allows the principal components to be based on a waveform shape rather than amplitude. We automatically identified clusters by fitting a mixture of Gaussians to the first five dimensions of principal components analysis space and checked and combined clusters and corresponding waveforms manually. Clusters were classified as single neurons based on (1) the inspection of the inter-spike interval histogram, (2) the consistency of waveform over time, and (3) if their Isolation distance, which is a measure of the separability between clusters and background activity, exceeded 15[36]. Any remaining threshold crossings were classified as a multi-unit activity. A total of 150 single neurons (74, 30, and 46 from monkeys 1, 2, and 3, respectively) and 240 multi-unit (113, 49, 78 from monkeys 1, 2, and 3, respectively) neuronal clusters were recorded. The results obtained for single- and multi-units were not significantly different; so, throughout the article, we refer to them as neurons and report the collective results.

Throughout the paper, we assess trends in our full dataset of 390 neurons; however, not all neurons were orientation selective for all stimuli, preventing within-group comparisons. Therefore, in Fig. 6, we focus on a sample of 50 neurons, which were highly orientation selective in all luminance and contrast conditions, allowing a within-group comparison.

**Visual stimulation**. Visual stimuli were generated using MATLAB with Psychtoolbox[52] and presented on an LCD monitor (Display + +, Cambridge Research Systems, UK) with 120 Hz refresh rate, a display width of 700 mm, a resolution of 1920 ×1080 pixels and a viewing distance of 500 or 700 mm. The monitor uses a built-in gamma-correction mechanism and has a 10-bit colour resolution[36]. Stimuli were viewed monocularly through the contralateral eye. Orientation selectivity was initially characterised using static gratings presented for 50 ms, followed by a grey blank screen for 500 ms. The gratings had 12 equally spaced orientations spanning 0–180°, six spatial frequencies (0.05, 0.125, 0.25, 0.5, 1, 2 cycles per degree) and two phases (0–180°). Spiking rates were averaged 50–150 ms after stimulus onset. We also obtained the contrast-response function of neurons using an optimal gating presented with different contrasts (4, 8, 16, 32, 64, and 100%).

**Luminance–contrast switching paradigm**. To study the dynamics of adaptation and how orientation selectivity is affected by luminance and contrast adaptation, we presented a movie of rapidly changing gratings with random orientations, phases, and spatial frequencies. Stimuli were full-screen sinusoidal gratings presented for two monitor frames (16.7 ms), with 12 equally spaced orientations (0–165°) and eight phases (0–360°). We selected one to three spatial frequencies in a range of 0.125–0.4 cycles per degree based on the spatial frequency tuning of recorded neurons in each penetration. Every 5 s, the luminance and/or contrast of gratings were randomly selected from four luminance–contrast combinations: (1) high contrast (65%), low mean luminance (70 cd m$^{-2}$); (2) low contrast (35%), low mean luminance (70 cd m$^{-2}$); (3) high contrast (65%), high mean luminance

(140 cd m$^{-2}$); and (4) low contrast (35%), high mean luminance (140 cd m$^{-2}$) (Fig. 1). This led to 12 different switch conditions, in which either or both luminance and contrast changed. The total presentation time was 60 min, yielding an average of 56 repetitions for each unique luminance–contrast switch. As luminance–contrast switching was random, the four no-change conditions also occurred with equal probability, but these are not analysed here. The contrast and luminance levels were selected so that most neurons were both responsive and orientation selective.

**Linear-response estimation (reverse-correlation analysis).** We characterised the functional properties of neural responses to variations in stimulus orientation using the orientation reverse-correlation method[18]. We computed reverse correlograms for each stimulus orientation by correlating the occurrence of each orientation with the spike train. Firstly, an array of counters for each of the 12 orientations and 512 time delays ($\tau$ = -6 to + 250 ms with 0.5 ms resolution) was constructed, $R(\theta,t)$, with all initial values set to zero. For each spike time, we looked $\tau$ earlier in time and incremented the counter corresponding to the presented stimulus orientation ($\theta$), regardless of grating phase. At the end of this procedure, the sum of the counters at each time delay was equal to the number of spikes collected. The resulting counts were normalised at each time delay, giving the relative probability that a spike was preceded by each possible orientation,$p(\theta, \tau)$.

We subsequently calculated several response characteristics for each neuron, including peak time (when the orientation selectivity was maximised), the maximum modulation (the difference between the highest and lowest probability at peak time), and the preferred orientation ($\theta_{pref}$), which evokes the highest probability ($p_{max}$). The preferred orientation ($\theta_{pref}$) was found by fitting the difference between two von Mises functions to the tuning curve at the peak time, Eq. (1). Taking the difference between two von Mises functions with different parameters allowed a better fit to asymmetric tuning functions.

$$p(\theta) = \alpha_1 e^{k_1 \cos(\theta - \theta_{pref\,1})} - \alpha_2 e^{k_2 \cos(\theta - \theta_{pref\,2})} + \beta \qquad (1)$$

where $\theta_{prefi}$ is the preferred orientation, $k_i$ is the width parameter, $a_i$ is the scaling factor and $\beta$ is a constant offset. We also extracted the temporal width, which is the time window over which the probability of the preferred orientation, $p(\theta_{pref}, t)$, exceeded half the maximum.

For most analyses, we calculated reverse correlograms in early (0–1.6 s) and late (3.4–5 s) time windows following each luminance–contrast switch. In total, this gives us 24 reverse correlograms (12 switches; 2 time windows). Our estimation of linear responses during these time windows was robust as we had over 350 repetitions of each unique stimulus orientation. For some analyses (Supplementary Fig. 1), we applied the reverse correlation method to the whole 5 s duration after the switch.

**Inclusion criteria.** We analysed those neurons that showed a significant orientation selectivity. Neurons were deemed to be orientation selective if they satisfied the three following criteria: (1) maximum probability (probability at peak time) in the reverse-correlation analysis exceeded 3 × sd of the baseline probability; (2) the von Mises function fit (Eq. 1) was significantly better than a flat line ($F$ test, $p < 0.05$); and (3) the bandwidth of von Mises function fit was 20–95°. Across all the recordings (511 isolated neurons), 453 neurons were visually responsive, and, of these, 390 neurons (86%) were orientation selective when tested with at least one of the luminance–contrast combinations. A neuron was selected as visually responsive if its spiking activity significantly increased above baseline after stimulus presentation based on the standard forward-correlation stimuli.

**Firing rate and trial-to-trial variability.** We calculated peri-stimulus time histograms (PSTHs) for every neuron from 5 s before to 5 s after each luminance–contrast switch. For simplicity, we disregard switches when both luminance and contrast simultaneously changed. PSTHs were calculated using 50 ms time bins and we ignored the variations in other stimulus dimensions (i.e., orientation, phase, and spatial frequency). Each PSTH was normalised relative to a shuffled PSTH, generated by shuffling the spike times. The shuffling process was repeated 50 times while the spike times across the entire 1 h dataset were randomised in each run. The switching PSTHs were normalised relative to the averaged shuffled PSTH. For the sake of visualisation, the resulted PSTHs were then convolved with a Gaussian window with 50 ms width.

To assess trial-to-trial variability, we calculated the FF, which is the ratio of variance to mean spike count across the trials. FF was calculated in a sliding window of 50 ms with a time step of 10 ms.

**Information theoretic analysis.** We quantified the amount of information conveyed by neurons about stimulus orientation using information-theoretic analysis. Therefore, the MI between the spiking activity and stimulus orientation was calculated using Eqs. (2) and (3):

$$MI_{t,\delta_t(\theta,R)} = \sum_{\forall \theta} p(\theta) \sum_{\forall R} p(R|\theta) \log_2\left(\frac{p(R|\theta)}{p(R)}\right) \qquad (2)$$

Where:

$$p(R) = \sum_{\forall R} p(\theta)p(R|\theta) \qquad (3)$$

Where, $R$ is the spike count in a time interval $\tau$ after grating onset of width $\delta t$, $p(\theta)$ is the probability of presenting orientation $\theta$, which is close to uniform in our case (1/12), $p(R)$ is the probability of observing response $R$ evoked across all stimuli, $p(R|\theta)$ is the conditional probability of observing response $R$ given grating with an orientation $\theta$ was presented. We also applied a bootstrap-based bias-correction method to have an unbiased estimation of information[53].

The above calculation was performed three times: (1) during the early-adaptation phase, which is the time window between 0 and 1.6 s after the switch (Fig. 1); (2) during the late-adaptation phase, which is the time window between 3.4 and 5 s after the switch (Fig. 1); and (3) during the course of adaptation in six non-overlapping time windows of 833 ms (Supplementary Fig. 4).

**Population-decoding analysis.** To estimate what information can be extracted about the stimulus orientation from a given neural population, we applied simple linear decoders to our dataset. Such simple decoders are biologically plausible as they perform classifications by computing the weighted sum of spike counts. The weights can be considered as the synaptic strength and the outputs of classifiers, which are based on a decision boundary, are analogous to neuron's spiking threshold. The main difference between the decoders is the way in which optimal weights and the decision boundary are learned. Here, we employed two simple and widely used decoders, including linear discriminant analysis (LDA) and support vector machines (SVM with linear kernel). The advantage of the support vector machine classifier is that it learns the structure of the neuronal response distributions without any particular pre-assumptions about the response distributions. These linear classifiers exhibit good performance, generalisation, and minimal overfitting for a wide range of neural data and applications[54,55].

Response matrix: before applying the decoder, we constructed a response matrix, **R**, which is an $S \times N \times T$ matrix, in which $S$ is the number of unique stimulus orientations, $N$ is the number of neurons, and $T$ is the number of trials for each stimulus. Each element of the matrix is the number of spike events elicited by each stimulus in each neuron over a given time window (e.g., 15 ms). Finally, the spike counts across neural populations ($N$) were normalised ($z$ score). This normalisation was done to compensate for spike-count variations in response to a stimulus across neural populations. The normalisation did not have any effect on the pattern of decoding accuracy and a small effect on absolute accuracy (∼1% increment). The response matrix, **R**, was built across multiple time windows starting from 0 to 200 ms after stimulus onset (2 ms temporal resolution). The response matrix, **R**, was calculated across different time windows during adaptation (after luminance–contrast switch; Supplementary Fig. 4). In some analyses (Supplementary Figs. 4 and 5), we varied different dimensions of the response matrix **R** and studied these changes on decoding accuracy.

Decoding analysis: in our analysis, we used 70% of the response matrix **R** to train the classifiers and 30% of the remaining to test. We separately trained and tested the decoder at early and late time periods after a stimulus switch and additionally performed separate training and testing for each type of stimulus switch. All the reported accuracies are the results of 15 cross-validated runs. The trials, $T$, and neurons from three animals, $N$, were randomly selected at every time delay, removing the effect of spike-count correlations. Here, all results are based on resampled subpopulations of 50 neurons. To statistically test whether a given mean decoding accuracy was significantly higher than chance, we repeated the same decoding procedure but shuffled the trial labels across trials and time.

**Multiplicative and additive modulations.** To examine the changes in tuning throughout the 5 s adaptation period, we first calculated PSTHs in a time window of 0–200 ms following the appearance of each orientation. Then, tuning functions were created by averaging the spike rate in a 20 ms time window centred on the time of the peak response to the preferred orientation. These tuning functions were calculated separately for gratings presented early (0–1.6 s) and late (3.4–5 s) after each luminance–contrast switch, giving a total of 16 tuning curves (Fig. 6; eight switches, two time periods). For each neuron and switch type, we characterised its multiplicative and additive modulations by performing linear regression on the average responses to each orientation, during the early phase of adaptation compared to the late phase. The slope of the linear fit indicates how tuning scales multiplicatively (gain modulation of tuning function), whereas the intercept of the fit describes the additive shift (offset). Thus, the responses of the neurons with a purely multiplicative modulation can be fit with a line that passes through the origin, while responses of neurons with a purely additive modulation can be fit by a line with a slope of one and a non-zero intercept.

We examined the structure of multiplicative and additive modulations of tuning curves of every neuron during early and late phases of different adaptation conditions using four models: (1) a model with a single Von Mises function, applied to both early and late phases; (2) a model with two independent Von Mises functions, one for the early and one for the late phase; (3) a model with two Von Mises functions, with the same gain for tuning curves at early and late phases but with different offsets; and (4) a model with two Von Mises functions, with the same

offset for tuning curves at early and late phases but different gains. To compare the performance of different models, we calculated the mean square error between the fit and experimental tuning curves. As the mean square error does not take into account the number of parameters in each model, we also calculated the Akaike's Information Criterion to show which model is the best predictor considering the number of parameters in each model.

**Human psychophysical experiments.** In order to investigate the perceptual correlate of orientation discrimination in cortical responses during adaptation to luminance and contrast, we performed psychophysical measurements in human observers. The stimulus design was very similar to the switching-stimulus paradigm in our monkey electrophysiology experiment, but here the subjects performed an orientation-discrimination task following adaptation to a period of continuously changing gratings.

On each trial, the observers fixated a black cross at the centre of the screen while a movie of rapidly changing gratings with random orientations and phases was presented. Stimuli were sinusoidal gratings with circular apertures of diameter 7°, centred on the fixation cross, and were presented for two monitor frames (16.7 ms), with 12 equally spaced orientations and eight phases (Fig. 5a). After 6 s of presentation, the contrast (luminance) of gratings was switched to a higher (lower) level. Either 0.2–1.2 s (early trials) or 5–6 s (late trials) after the switch to the new contrast (luminance), a test grating was shown, followed by dynamic filtered oriented noise[56], and a second test grating, each presented for 200 ms (Fig. 5a). Test gratings and the dynamic noise mask had the same luminance and contrast. Finally, observers indicated with a keyboard press whether the second test grating was oriented clockwise or counter clockwise relative to the first test grating. The stimulus timings used for the psychophysical tests differed slightly from those in the physiological study. To increase any possible adaptation effects, we used slightly longer adaptation periods, coupled with a larger relative delay between the early and late periods. To minimise any distracting or masking effect associated with the period immediately after a luminance or contrast switch, we also delayed the start time of when the test gratings could appear to be 0.2 s after the switch.

In the contrast-switch task, the contrast switched between 10 and 100%. In the luminance-switch task, the grating (and background) luminance switched between 0.1 and 0.9 (normalised luminance) while having 10% contrast throughout the trial. We used the method of constant stimuli with 14 levels of orientation difference, ΔOri (seven clockwise), with logarithmic intervals. The sampling resolution varied depending on subject's threshold during practice trials. Auditory feedback at the end of each trial indicated correct and incorrect responses. As a control, we measured the subjects' orientation discrimination threshold at low (10%) and high (100%) contrast as well as low and high luminance levels. These trials consisted of two test gratings, which were temporally separated by a filtered oriented noise, but with no preceding adaptation gratings. In addition, in some trials, the discrimination task was shown 2–3 s into the trial. We did not analyse these trials and only included them for observers' vigilance.

Stimuli were generated using MATLAB with Psychtoolbox[52] and were presented on an LCD monitor (ViewPixx 3D; VPixx Technology Inc., Saint-Bruno, QC, Canada) with 120 Hz refresh rate, a diagonal display size of 22.5″, the maximum luminance of 106 cd m$^{-2}$, a resolution of $1920 \times 1080$ pixels, and a viewing distance 60 cm. We gamma corrected the monitor and used a 10-bit colour resolution. The experiments were performed in a dark room while subjects were comfortably seated on a chair and their head was rested on a chinrest. Experiments were completed in two sessions lasting 60–90 min, with each trial condition presented at least 15 times.

We calculated the ratio of clockwise responses at every ΔOri level for each trial condition separately and fit a cumulative Gaussian function to data using maximum likelihood methods. The discrimination thresholds (measured as just noticeable difference) were then compared at different conditions. Non-parametric statistical tests were performed, based on boot-strapped resampling of each participant's data 1000 times with replacement.

In total, we recorded the behavioural data from 31 subjects (16 females, aged 20–34 years, with normal or corrected-to-normal vision). The data of two subjects were excluded because their psychometric curves were flat at 50%, indicating that they were guessing or did not understand the task. Note that some subjects participated in more than one of the four luminance and contrast tasks. Subjects were students from the Faculty of Medicine, Nursing and Health Sciences, Monash University. All subjects voluntarily participated in the experiments and gave their written consent prior to participation. All human psychophysical experiments were conducted in accordance with the National Statement on Ethical Conduct in Human Research and all procedures were approved by the Monash University Human Research Ethics Committee.

**Reporting summary.** Further information on experimental design is available in the Nature Research Reporting Summary linked to this article.

## Data availability
The datasets for the current study are available from the corresponding author on request. MATLAB code used for the analysis is available from the corresponding author on request.

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

## Acknowledgements

We thank Richard Born and Nathan Crowder for their comments on the manuscript. This work was supported by the National Health and Medical Research Council (APP1066588 and APP1120667); the Human Frontier Science Program Career Development Award to NSCP; the Australian Research Council Special Research Initiative in Bionic Vision; and the Australian Research Council Centre of Excellence for Integrative Brain Function. We thank Janssen-Cilag Pty Limited for the donation of sufentanil citrate.

## Author contributions

M.G. and N.P. designed the experiments. M.G., E.Z., N.P and M.R. performed the experiments. M.G. and N.P. analysed the data. M.G., N.P and E.Z. wrote the manuscript.

## Additional information

**Competing interests:** The authors declare no competing interests.

**Journal Peer Review Information**: *Nature Communications* thanks the anonymous reviewer(s) for their contribution to the peer review of this work. Peer reviewer reports are available.

