## [Peer Review File · Nature Communications]

Reviewers' comments:

Reviewer #1 (Remarks to the Author):

One of the key mysteries in sensory neuroscience is how neurons can maintain a constant code for a sensory variable in the external world (such as orientation) in the face of known changes in neuronal responsivity due to other, 'nuisance' features of the visual world that are irrelevant to that variable (such as luminance or contrast changes). This study asks, using population analysis of recordings in V1 from anesthetized marmosets combined with human psychophysics and computational modeling, how the orientation code is affected by these two 'nuisance' variables (luminance and contrast). They find that orientation discriminability is affected by luminance and contrast, and propose that adaptation to these variables serves to keep the visual system operating within its limited dynamic range.

Overall the study is well written, and the topic is of broad interest with a well-posed question. I found the analyses to be rigorous overall, and the results seem robust. The difficulty I have stems from the interpretational framework applied to the mix of neurophysiology, computational, and psychophysical results. The link between the neurophysiology and psychophysics is tenuous, and the model doesn't really add much. Despite the strength that comes from a great set of neurophysiological data and strong computational analyses that have merit on their own as reportable, novel scientific findings, the package of results here does not stand well as a coherent set of work supporting the overall thesis of the authors. This is primarily due to some mismatch between the psychophysics, modeling, and neurophysiology.

Specific Comments:

- In the psychophysical study, the observers could discriminate better after an increase in contrast (this part was expected) and after a luminance decrement (which seems surprising). No studies were done using a luminance increment or contrast decrement, which is a shame as it would be good to see the lack of improvement for the luminance increment.
- Were pupil diameters monitored in the psychophysical study? I see no mention of eye tracking. When the luminance decreased presumably pupils dilated. This wouldn't be possible in the anesthetized animals where dilating drops were used. This makes it essentially impossible to compare the psychophysics and physiology without tracking data to know what happened to the pupils. This seems like a critical flaw in the comparison.
- The early/late period from the psychophysical study (0.2-1.2 s vs 5-6 s) and data analysis (0-1.6 s vs 3.4-5 s) aren't matched. Why?
- The model exercise showed that a rate-based E/I-coupled model could match their results, particularly if two separate inhibitory populations were included. The citation to the Litwin-Kumar et al manuscript (#57) isn't appropriate here since they use a spiking network model, which is different than the fitted rate-based model of the authors. Given that the model is fit to the data to produce the results shown, what is exactly the strength and predictive power supplied by the model? It's already well known that recurrent network models can reproduce a large number of phenomenon, and with the available parameters here and the fitting to data it would actually surprise me if it couldn't show the neurophysiological effects they find.
- The study only analyzed neurons that showed significant orientation selectivity and were visually responsive. While this seems like it makes some sense given that the authors are studying orientation

coding, it is unclear that the readout of V1 could selectively access tuned cells. It would be worthwhile to run the key analyses with all neurons included to see if it changed any key results. Given that decoding approaches would simply down-weight information from untuned neurons, and weakly tuned neurons would improve the code, it's strange the authors chose to eliminate any neurons. What could matter, however, is if untuned neurons tended to show more non-stationarity across luminance or contrast changes.

- An overarching issue that's not addressed is that luminance and contrast are only nuisances from the perspective of a decoder that wants to know about orientation, but from the perspective of the animal they are alternative stimulus variables. The system does actually have information about the true luminance and contrast on some timescales (i.e., it would be decodable). The authors have a clear goal for their study which focuses on orientation, which is fine and appropriate given what we know about luminance/contrast coding in the earliest stages of visual processing, but this issue should at least be discussed.

Additional (minor) comments:

- No mention is made of the number of neurons that are dropped for a lack of visual responsiveness (only the % of visually responsive neurons that met the orientation tuning criterion)

- p. 26, "optimal gating" typo should be "an optimal grating"

- In the methods they state they use LDA and SVM. One figure legend mentions where LDA is used. Where is SVM used? There needs to be more clarity about the particular decoding analysis used in each situation.

- A mask is used in the psychophysics. What contrast and properties did it have?

Reviewer #2 (Remarks to the Author):

This manuscript describes the effects of luminance and contrast adaptation on the coding of orientation in V1 of the marmoset. Luminance increments and contrast decrements lower firing rates and hence reduce stimulus information in the population. Firing rates and information then recover over several seconds, but the recovery is incomplete.

Although these are interesting new data, much of the paper focusses on things that are already quite well understood. Many groups have measured orientation tuning at different contrast levels. These past studies have all been in the steady state, after contrast adaptation/gain control is complete. But these existing measures are essentially equivalent to the findings reported here for the late conditions. Although they have added some new quantifications, there is nothing surprising here, as far as I can tell. So they need to make a clearer case about what is and is not novel.

So the really novel data here are the measurements of orientation tuning during adaptation, where using reverse correlation allows them to follow the timecourse not just of mean firing rate, but of the whole response. To some extent what they report here is also unsurprising (e.g. a contrast decrement initially reduces tuning, prior to adaptation), but it's the first time its been measured. One of the interesting findings here is that during this period the tuning curve is multiplicatively scaled. Given the novelty of this finding, it might be worth pursuing in more detail. Currently they only compare gain in two time periods. Do they have enough data to measure the gain with finer temporal resolution and

show us a timecourse for the gain, analogous to the timecourse currently shown for rates? Is the contribution of gain and offset constant during the evolving part of the response?

The modelling section adds relatively little. In particular, the fact that they can build a recurrent cortical model that matches the data does not exclude other possibilities. For example, it could be that all of the contrast gain control and luminance adaptation takes place in the LGN, and the effects are inherited in cortex. I don't see any clear conclusion from the modelling.

Smaller points.

The figure legends could be more helpful. The color scheme in Figs 2 and 3 (e.g the dark/light lines in Fig 2D) are not explained in the legend. I had to hunt the main text for this. In fig 4a, its not very clear what the r values refer to. The correlations were calculated across the 6 time windows, within each condition, but it took me a while to figure this out.

"our results revealed that an increment in contrast can trigger a significant change in the shape of the linear kernel during the early phase of adaptation compared to the late phase, suggesting a gradual increase in the amplitude of the kernel during contrast adaptation. However, during a contrast decrement, the kernel amplitude remained largely unaffected during adaptation." Something must be missing here. As written, it suggests that a contrast increment followed by a contrast decrement, leaves the system in an altered state, even though contrast is not back to its initial value.

The manuscript starts by using reverse correlation, but then changes to using forward correlation when measuring response gains. Might it be simpler to use forward correlation throughout? Also, counting spikes from 0 to 200ms seems like a very long window for a 16ms stimulus. Can't they use rates at the peak response time?

I am puzzled by the population decoding, which starts with an $S \times N \times T$ matrix. This requires that all N neurons be recorded simultaneously. Was this decoding done separately for each recording session? Does the population decoding reveal anything that is not seen at the single neuron level? It seems like both analyses give the same result. Which is fine, but its not made clear for the reader that these are just two descriptions of the same phenomenon.

Reviewer #3 (Remarks to the Author):

This manuscript uses a variety of techniques to examine the interaction between neural adaptation and sensory coding, and the concomitant impact these interactions have on population coding and perception. Specifically, they find that the orientation discrimination is modulated by seemingly orthogonal factors, such as stimulus contrast and luminance —an effect attributed to changes in gain of the response. The study is tied together nicely with a simple, yet, elegant proof of concept psychophysical study that support their marmoset findings, as well as a recurrent gain control model. There's quite a bit to like about this study, warranting publication; the methods are rigorous and thoughtful and the techniques are considered at multiple scales of theoretical importance. Below I outline my comments and concerns, which frankly for this study are few and far between!

1) Perhaps this is just semantics, but the reference to contrast and luminance as "nuisance variables" both in the title and main manuscript seems odd. Particularly since the point of the study is to reveal that they are anything but nuisance variables. But also because they are usually considered stimulus parameters of main interest in many other studies. The authors might want to reconsider that framing.

2) While adaptation is not a major component of that work, the authors should consider making reference to previous work by authors such as David Ferster and Robert Shapley examining the contrast-invariant relationship between contrast and orientation tuning.

3) In the methods, the authors state that contrast response functions were assessed per neuron, spanning a nice range of contrasts. Was this done at different luminances as well? The outcome of the contrast (and luminance) increment/decrement manipulation might be expected to vary depending on the contrast sensitivity profile for a given neuron. For instance, if the CRF had high sensitivity, there may be little headroom for detectable changes in response to increments rather than decrements, simply due to the saturating nonlinearity. Same goes with luminance response. If the cortical luminance response function had plateaued given the base luminance tested, one might expect a increment/decrement asymmetry simply due to that nonlinear profile. Perhaps the authors could examine this dependency. It would be equally fascinating if the switch effects were independent of the contrast sensitivity of a neuron.

4) Some of the figures could benefit from legends (such as Figure 3), as it was sometimes difficult to decipher what condition a given trace corresponded to.

Response to reviewer comments

We would like to thank the reviewers for their useful comments and criticisms.

We have made extensive changes throughout the manuscript, including the addition of new human psychophysical data (addressing Reviewer 1's comments) and new analyses to examine how adaptation causes multiplicative scaling of neuronal responses in the physiological data (addressing Reviewer 2's comments). We believe these changes link the physiology, modelling and psychophysics more coherently, and expand the novelty and importance of the work. These improvements required substantial changes to figures and text throughout the manuscript.

Additionally, both Reviewers 1 and 2 raised doubts about whether the modelling was necessary. While it is a relatively small portion of the manuscript, its importance is to identify a mechanism or circuit that could account for our data. Our results show that a relatively simple and biologically plausible model can account for the complexity of the physiological observations. To make this contribution clear, we have added text describing that the computational modeling showed that a simple gain control model that has successfully been used to predict the adaptation in the retina and LGN is *not* sufficient to accurately predict gain modulations in V1; however, a recurrent network model incorporating excitatory and multiple inhibitory neural populations can predict the detailed time-course of experimentally observed modulations. We believe these changes contextualize the model, and clarify its role in the manuscript's narrative.

Below, the **reviewer's comments (or summaries) are in bold**, and our responses are in blue.

Reviewer #1 (Remarks to the Author):

In the psychophysical study, the observers could discriminate better after an increase in contrast (this part was expected) and after a luminance decrement (which seems surprising). No studies were done using a luminance increment or contrast decrement, which is a shame as it would be good to see the lack of improvement for the luminance increment.

We agree with the reviewer that including the additional psychophysical studies is an important way to increase the consistency of the physiology and human psychophysics.

In the earlier version of the manuscript, we only reported psychophysical data for those conditions in which we observed a strong effect in the physiological data. Notably, contrast increments and luminance decrements were associated with the largest changes in spiking rate (Fig. 3 BC), and effects on decoding and mutual information. To address the reviewer's concerns, we tested 15 additional participants to determine how luminance increments and contrast decrements affected orientation discrimination. As the reviewer predicted, we did not observe any systematic, or significant, changes in discrimination threshold in the early compared to late phase (Figure 5, D G, page 16). While these further experiments produced no effect, we agree with the reviewer that it is an important null result, and in the context of the paper, helps to link the physiology and psychophysics together.

Were pupil diameters monitored in the psychophysical study? I see no mention of eye tracking. When the luminance decreased presumably pupils dilated. This wouldn't be possible in the anesthetized animals where dilating drops were used. This makes it essentially impossible to compare the psychophysics and physiology without tracking data to know what happened to the pupils. This seems like a critical flaw in the comparison.

As outlined in our previous response, we conducted additional psychophysical experiments to examine the effects of luminance-increments and contrast-decrements on orientation discrimination. For all 15

observers performing these experiments, we also monitored eye position and pupil size. In addition, we collected pupil size data from a single naïve observer who repeated the original contrast-increment and luminance-decrement studies in order to perform pupillometry. As expected, pupil size did not change substantially after contrast switches, but rapidly constricted after a luminance increment and more slowly dilated after a luminance decrement (Supp. Fig. 6 in the manuscript, also copied below). The dynamic variations in pupil size largely occurred during the earlier of the two possible analysis windows. The difference in pupillary dynamics following luminance and contrast switches highlights that they alone cannot account for our observed psychophysical results: similar improvements in perceptual thresholds were evident following both contrast increments and luminance decrements, but only one of these stimulus conditions was associated with changes in pupil diameter. Note also that the discrimination threshold was significantly different between the early and late phases following a contrast increment switch, while the pupil size was mostly unaffected during the switch.

Figure S6. Modulation of pupil size during luminance and contrast switches. Related to Figure 5. (A-D) Each trace shows the change in pupil area following a luminance or contrast switch. While luminance switches strongly affect pupil size, contrast switches had minimal effects. The area specified by “task window” is the time window when subjects performed the orientation discrimination task. The traces show the mean (SEM) of pupil area, averaged across all trials for all observers. Pupil size for each observer was normalized to have zero mean and unit standard deviation (z score) across all trials. Number of participants from A-D: n=7, 1, 1, 8.

The early/late period from the psychophysical study (0.2-1.2 s vs 5-6 s) and data analysis (0-1.6 s vs 3.4-5 s) aren't matched. Why?

We agree with the reviewer that this difference might seem strange. However, our aim with the psychophysical experiments was to show *qualitative* agreement between the nature of changes in human orientation discrimination and those predicted by our physiological data, rather than to exactly or quantitatively replicate the nature of the stimulus presentation. In fact, only a qualitative agreement is possible because for the human psychophysical study, observers discriminated the relative orientation of

two briefly presented gratings after a period of adaptation, whereas for the physiological data, we decoded the orientation of continuously-presented gratings with changing orientation.

The differences in stimulus protocol are relatively minor, and we have added text explaining our rationale for their selection (page 39, Methods):

“The stimulus timings used for the psychophysical tests differed slightly from those in the physiological study. To increase any possible adaptation effects, we used slightly longer adaptation periods, coupled with a larger relative delay between the early and late periods. To minimise any distracting or masking effects associated with the period immediately after a luminance or contrast switch, we also delayed the start time of when the test gratings could appear to be 0.2 s after the switch.”

The model exercise showed that a rate-based E/I-coupled model could match their results, particularly if two separate inhibitory populations were included. The citation to the Litwin-Kumar et al manuscript (#57) isn't appropriate here since they use a spiking network model, which is different than the fitted rate-based model of the authors. Given that the model is fit to the data to produce the results shown, what is exactly the strength and predictive power supplied by the model? It's already well known that recurrent network models can reproduce a large number of phenomenon, and with the available parameters here and the fitting to data it would actually surprise me if it couldn't show the neurophysiological effects they find.

We agree that it would be surprising if a sufficiently-complex recurrent-network model couldn't reproduce our observed physiological data. However, we believe that it is important to demonstrate that a relatively simple, and biologically plausible model can capture the complex array of effects observed with electrophysiology. In response to the reviewer comment, we have revised the wording throughout the Modeling section of the Results and added further explanation of the merits of the modeling in the Introduction and Discussion.

Page 3, Introduction (last paragraph):

“These changes in gain are largely predicted by an extremely simple rate-based gain control model, however, the detailed time-course of these modulations could only be explained by a model incorporating recurrently connected excitatory and multiple types of inhibitory sub-populations of neurons, which receive feed-forward inputs that are intrinsically luminance- and contrast-dependent.”

Page 22, Discussion (end of first paragraph):

“Our computational modeling showed that a simple gain control model that has successfully been used to predict the adaptation in the retina and LGN is not sufficient to accurately predict gain modulations in V1; however, a recurrent network model incorporating excitatory and multiple inhibitory neural populations can predict the detailed time-course of experimentally observed modulations.”

We have removed the citation to Litwin-Kumar et al.

The study only analyzed neurons that showed significant orientation selectivity and were visually responsive. While this seems like it makes some sense given that the authors are studying orientation coding, it is unclear that the readout of V1 could selectively access tuned cells. It would be worthwhile to run the key analyses with all neurons included to see if it changed any key results. Given that decoding approaches would simply down-weight information from untuned neurons, and weakly tuned neurons would improve the code, it's strange the authors chose to eliminate any neurons. What could matter, however, is if untuned neurons tended to show more non-stationarity across luminance or contrast changes.

We agree that it is unclear that the readout of V1 could selectively access tuned cells, and that decoding approaches should down-weight information from untuned or poorly tuned neurons. Our rejection criteria are not particularly restrictive, and include many weakly tuned neurons already. Untuned neurons can only influence decoder performance if structured noise correlations exist in the population. Similarly, weakly tuned neurons will only improve decoder performance if they are strongly correlated with the highly tuned neurons.

However, in the decoding analyses presented here, the trials and neurons from three animals were randomly selected at every time delay. This was necessary to achieve sufficient data sets for training and testing the decoders, but has the effect of removing spike-count correlations. Therefore, incorporating the untuned neurons cannot systematically change the results of our decoding analyses, as expected given previous studies by us (Zavitz et al. 2016) and others (Leavitt et al. 2017, PNAS; Zylberberg 2018, BioRxiv).

To address the reviewer's concerns, we have expanded the description of the decoding in the Methods (page 33):

"We separately trained and tested the decoder at early and late time periods after a stimulus switch, and additionally performed separate training and testing for each type of stimulus switch. All the reported accuracies are the results of 20 cross-validated runs. The trials, T, and neurons from three animals, N, were randomly selected at every time delay, removing the effect of spike-count correlations. Here, all results are based on resampled subpopulations of 50 neurons. To statistically test whether a given mean decoding accuracy was significantly higher than chance, we repeated the same decoding procedure but using shuffled the trial labels across trials and time."

An overarching issue that's not addressed is that luminance and contrast are only nuisances from the perspective of a decoder that wants to know about orientation, but from the perspective of the animal they are alternative stimulus variables. The system does actually have information about the true luminance and contrast on some timescales (i.e., it would be decodable). The authors have a clear goal for their study which focuses on orientation, which is fine and appropriate given what we know about luminance/contrast coding in the earliest stages of visual processing, but this issue should at least be discussed.

We agree with the Reviewer's point that luminance and contrast are encoded within the visual system, and are specifically defined as nuisance variables here only because we focus on the encoding of orientation. We have addressed this by adding statements in the Introduction and Discussion.

We now establish our treatment of contrast and luminance with respect to our decoding paradigm early on (page 3, Introduction):

"Our study is the first to reveal how orientation coding in V1 neurons is impacted by adaptation to presumably orthogonal stimulus dimensions. Note that while encoding luminance and contrast is a critical aspect of the visual system, for the specific case of orientation-encoding, we treat them here as nuisance variables."

We also discuss how this perspective on luminance and contrast may be carried forward to future work, conducted from a different perspective (page 25, Discussion):

"The aim of our study was to explore the effect of adaptation to nuisance variables, such as luminance and contrast, on the coding of stimulus orientation. Clearly, luminance and contrast are only problematic from the perspective of encoding stimulus orientation - neurons in most cortical areas actually carry information about luminance and contrast, and this information can be reliably decoded. While not the focus of our study, this is evident in our data as difference in the steady-state firing rate after adaptation to luminance and contrast (e.g. the mean firing rates for low and high contrasts are different and thus decodable)."

Additional (minor) comments:

- No mention is made of the number of neurons that are dropped for a lack of visual responsiveness (only the % of visually responsive neurons that met the orientation tuning criterion)

We have clarified the numbers of neurons in the Methods (p31):

“Across all recordings (511 isolated neurons), 453 neurons were visually responsive, and of these, 390 neurons (86%) were orientation selective when tested with at least one of the luminance-contrast combinations.”

- p. 26, “optimal gating” typo should be “an optimal grating”

Page 29: This has been corrected.

- In the methods they state they use LDA and SVM. One figure legend mentions where LDA is used. Where is SVM used? There needs to be more clarity about the particular decoding analysis used in each situation.

Our apologies for the confusion. We used both LDA and SVM decoders, but we only reported the results of LDA (as a simple linear classifier). The results obtained with the two decoders were very similar, and this has been clarified in the main text and the figure legends. The results for the original LDA classifier (left) and SVM classifier (right) are presented below. We would be happy to add them as a Supplementary Figure, but do not think it adds anything to the story, since we are interested in relative decoding performance, not finding an algorithm that provides the best possible decoding performance.

Res. Let. Figure 1. Decoding accuracies Early and Late following each of the 8 luminance-contrast switches. Each data point shows the average and standard deviation of decoding accuracy for one switching condition (averaged over 15 runs). Solid points indicate upward switches; empty points indicate downward switches. The inset demonstrates the colour code for different luminance and contrast switches (e.g., dark blue refers to contrast increment when gratings had low mean luminance). Decoding algorithm: (A) LDA and (B) SVM, number of neurons = 50 randomly selected out of 390 neurons, width of spike counting window, 15 ms, number of random runs = 15. Error bars are standard deviation.

- A mask is used in the psychophysics. What contrast and properties did it have?

The mask had the same luminance and contrast as the two test gratings. This information has been added to the manuscript (page 39): “Test gratings and the dynamic noise mask had the same luminance and contrast.”

Reviewer #2 (Remarks to the Author):

They need to make a clearer case about what is and is not novel. So the really novel data here are the measurements of orientation tuning during adaptation, where using reverse correlation allows them to follow the timecourse not just of mean firing rate, but of the whole response. To some extent what they report here is also unsurprising (e.g. a contrast decrement initially reduces tuning, prior to adaptation), but it's the first time its been measured.

We have emphasised the novelty of our approach, and the data, by rewriting multiple parts of the Introduction and Discussion.

To summarise, our manuscript takes a novel approach to adaptation in sensory neurons by asking *how does adaptation to one stimulus dimension affect the coding of another dimension?* Here, for the first time, we have shown that:

- the coding of stimulus orientation by single V1 neurons changes during adaptation to presumably orthogonal stimulus dimension, such as luminance and contrast;
- based on reverse correlation analysis, the temporal kernel of neurons' orientation tuning is rescaled during adaptation;
- following changes in luminance and contrast, the ability of single neurons and neural population to code stimulus orientation dynamically changes, and these changes predict a novel form of perceptual adaptation in human orientation discrimination thresholds.

We expect that our stimulus and analytical approaches will be broadly relevant to analogous problems of assessing how robust neuronal representations are formed and maintained in other sensory and perceptual systems.

One of the interesting findings here is that during this period the tuning curve is multiplicatively scaled. Given the novelty of this finding, it might be worth pursuing in more detail. Currently they only compare gain in two time periods. Do they have enough data to measure the gain with finer temporal resolution and show us a timecourse for the gain, analogous to the timecourse currently shown for rates? Is the contribution of gain and offset constant during the evolving part of the response?

As the reviewer suggests, there is an evolving, but systematic, modulation in gain and offset throughout the adaptation period. A new figure (Supp. Fig. 7, copied below) shows the changes in the population tuning curve across six consecutive non-overlapping time windows during adaptation following each type of switch. The following text describing this finer-timescale analysis has also been added to the bottom of the section *Additive and multiplicative modulation of orientation tuning explains stimulus coding during adaptation* (page 18):

“We also calculated the gain and offset modulation with a finer temporal resolution during adaptation. To this end, we calculated the changes in the population tuning curve across six consecutive non-overlapping time windows during adaptation following each type of switch (Figure S7). Our analysis showed that the gain and offset systematically change during adaptation (Figure S7). In particular, we

found that the modulations during adaptation are not purely multiplicative or adaptive, however, the multiplicative modulations are more pronounced across single neurons.”

Figure S7 Modulations of orientation tuning during adaptation. Related to Figure 6. The average population tuning function of 50 neurons over six consecutive non-overlapping time windows throughout the 5 s adaptation period. To calculate the population tuning curve at each time window, we first shifted the preferred orientation of each individual neuron to 0° and then averaged across all neurons. Each panel shows tuning curves following a single switch, as indicated by the insets on the right. We only considered highly orientation-selective neurons for these analyses ($n=50$). There was strong and systematic multiplicative scaling following luminance increments (A,E) and contrast decrements (D,H), and a combined effect of offset and gain modulations following contrast increments (C,G) and luminance decrements (B,F).

This analysis also highlights the different timescales of responses. Following contrast increments and luminance decrements, tuning curve modulations largely occur during the first several hundreds of milliseconds after the switch; following luminance increments and contrast decrements there was a more gradual change in the gain of tuning curves.

The modelling section adds relatively little. In particular, the fact that they can build a recurrent cortical model that matches the data does not exclude other possibilities. For example, it could be that all of the contrast gain control and luminance adaptation takes place in the LGN, and the effects are inherited in cortex. I don't see any clear conclusion from the modelling.

The response below is copied from the response to a similar question posed by Reviewer 1.

We agree that it would be surprising if a sufficiently-complex recurrent-network model couldn't reproduce our observed physiological data. However, we believe that it is important to demonstrate that a relatively simple, and biologically plausible model can capture the complex array of effects observed with electrophysiology. In response to the reviewer comment, we have revised the wording throughout the Modeling section of the Results and added further explanation of the merits of the modeling in the Introduction and Discussion.

Page 3, Introduction (last paragraph):

These changes in gain are largely predicted by an extremely simple rate-based gain control model, however, the detailed time-course of these modulations could only be explained by a model incorporating recurrently connected excitatory and multiple types of inhibitory sub-populations of neurons, which receive feed-forward inputs that are intrinsically luminance- and contrast-dependent.

Page 22, Discussion (end of first paragraph):

"Our computational modeling showed that a simple gain control model that has successfully been used to predict the adaptation in the retina and LGN is not sufficient to accurately predict gain modulations in V1; however, a recurrent network model incorporating excitatory and multiple inhibitory neural populations can predict the detailed time-course of experimentally observed modulations."

Smaller points.

The figure legends could be more helpful. The color scheme in Figs 2 and 3 (e.g the dark/light lines in Fig 2D) are not explained in the legend. I had to hunt the main text for this.

Our apologies. We have tried to make this clearer - although due to the number of conditions, it will never be simple! We have modified the legend of figures 2, 3, and 4 to include clearer schematics, and extended the explanation of the conditions in each figure legend.

In fig 4a, its not very clear what the r values refer to. The correlations were calculated across the 6 time windows, within each condition, but it took me a while to figure this out.

We have rewritten the legend for Figure 4 and added an inset that describes the color-codes: "Note that the correlations between firing rate and MI were calculated across the six time windows within each condition."

"Our results revealed that an increment in contrast can trigger a significant change in the shape of the linear kernel during the early phase of adaptation compared to the late phase, suggesting a gradual increase in the amplitude of the kernel during contrast adaptation. However, during a contrast decrement, the kernel amplitude remained largely unaffected during adaptation." Something must be missing here. As written, it suggests that a contrast increment followed by a contrast decrement, leaves the system in an altered state, even though contrast is [not?] back to its initial value.

We assumed that the word “not” was mistakenly included in the reviewer’s last sentence (copied above).

It was not our intention to imply that the neural effects of a contrast-increment and contrast-decrement are non-commutative. Clearly, after a contrast-increment followed by a contrast-decrement, the stimulus has returned to its original state and we expect that the neural state will also return to its original setting.

As is evident in the population firing rate (Fig. 3), the short- and long-term effects of contrast increments and decrements are qualitatively different, with contrast increments producing a strong initial increase in firing rate, which relaxes to a smaller, but sustained higher level. Conversely, contrast decrements produce a rapid, small decrease in firing rate, which does not change much over time. Critically, if the solid and dotted dark blue lines in Fig. 3B and D are compared, it is evident that the initial firing rate of one curve matches the final firing rate of the other curve.

We have rewritten the Discussion to avoid possible confusion (page 23):

“For example, our results revealed that an increment in contrast can trigger a significant change in the shape of the linear kernel during the early phase of adaptation compared to the late phase, suggesting a gradual increase in the amplitude of the kernel during adaptation to high contrast. However, following a contrast decrement, the kernel amplitude did not significantly change, likely because of the fact that changes in response amplitude after a contrast decrement are small and have a very slow time-course of recovery.”

The manuscript starts by using reverse correlation, but then changes to using forward correlation when measuring response gains. Might it be simpler to use forward correlation throughout?

Forward correlation is certainly simpler to explain, but reverse correlation is appealing as it allows us to track changes in the temporal dynamics of neurons’ tuning functions and kernels during adaptation. The rationale for reverse correlation is described on page 4 (Results):

“In order to quantify orientation tuning in fine temporal detail, we used a reverse correlation approach to estimate the probability that each possible orientation occurred at all times preceding an action potential¹⁸. Applying orientation reverse correlation in the context of adaptation to stimulus luminance and contrast allowed us to examine how adaptation impacts tuning over time.”

Also, counting spikes from 0 to 200 ms seems like a very long window for a 16 ms stimulus. Can’t they use rates at the peak response time?

For each orientation, we calculated the PSTH from 0 ms to 200 ms after stimulus onset. Critically, the tuning curves were extracted only from the time of the peak response in the PSTH (i.e. not from the full 200 ms window). To clarify this we, have changed the text in the Methods section (page 33):

“We first calculated the PSTHs in a time window 0-200 ms following the appearance of each orientation. Then, orientation tuning functions were extracted from PSTHs by averaging the spike rate at the peak response time.”

I am puzzled by the population decoding, which starts with an $S \times N \times T$ matrix. This requires that all N neurons be recorded simultaneously. Was this decoding done separately for each recording session?

We applied the decoder to a subset of 50 neurons randomly selected from 390 neurons recorded from three animals. While the recordings were performed with arrays, we did not record sufficient neurons in a single array implantation or penetration to decode “true” population data containing neurons with both diverse tuning and intact correlations. Instead, to benefit from the large number of neurons overall, the trials and neurons were randomly selected for each run of the decoder, which removes spike count

correlations. We have amended the manuscript, adding the description (page 33-13):

“We separately trained and tested the decoder at early and late time periods after a stimulus switch, and additionally performed separate training and testing for each type of stimulus switch. All the reported accuracies are the results of 20 cross-validated runs. The trials, T, and neurons from three animals, N, were randomly selected at every time delay, removing the effect of spike-count correlations. Here, all results are based on resampled subpopulations of 50 neurons.”

Does the population decoding reveal anything that is not seen at the single neuron level? It seems like both analyses give the same result. Which is fine, but it's not made clear for the reader that these are just two descriptions of the same phenomenon.

Individual neurons in our population have different adaptive properties. For example, the time constant (Supplementary Figure 3) and the strength of adaptation vary across single neurons. This affects their coding properties as shown, for example, by the range of properties seen in the Mutual Information analyses (Figure 2, 4).

The aim of the population decoding was to investigate whether we can track changes in the population code for stimulus orientation during adaptation *despite* the variations at single neuron level. This is now addressed when the population decoding results are first described (page 12-13):

“It is, however, unclear how the coding of orientation by a neural population is affected by changes in firing rate and trial-to-trial variability at the level of individual neurons, and whether the variability between neurons can be overcome at the population level. Therefore, we used a population decoding approach to measure the orientation discriminability afforded by neural populations, asking how decoding accuracy: (1) is affected by different luminance and contrast conditions; and (2) how it changes during the course of adaptation to a single luminance and contrast.”

Reviewer #3 (Remarks to the Author):

1) Perhaps this is just semantics, but the reference to contrast and luminance as “nuisance variables” both in the title and main manuscript seems odd. Particularly since the point of the study is to reveal that they are anything but nuisance variables. But also because they are usually considered stimulus parameters of main interest in many other studies. The authors might want to reconsider that framing.

We agree with the reviewer’s sentiment that luminance and contrast are anything but a nuisance for functional vision. We have chosen to continue to use the term based on its origins in probability theory, in which nuisance variables are parameters that affect a probabilistic model, but which are not of specific interest. This framework is appropriate for our study, because in the context of encoding or decoding orientation (and only orientation), luminance and contrast are not parameters of interest. To make this distinction clearer, we have added to the Introduction (page 3):

“Our study is the first to reveal how orientation coding in V1 neurons is impacted by adaptation to presumably orthogonal stimulus dimensions. Although the encoding of luminance and contrast is a critical function of the visual system, for the specific case of orientation-encoding, we treat them here as nuisance variables.”

2) The authors should consider making reference to previous work by authors such as David Ferster and Robert Shapley examining the contrast-invariant relationship between contrast and orientation tuning.

We now cite two additional papers in the first paragraph of the introduction: “Considering just contrast and orientation, two dimensions that profoundly affect the firing rates and response dynamics of neurons in the early visual system 1,2,....”

- 1- Sceniak, M. P., Hawken, M. J., & Shapley, R. (2002). Contrast-dependent changes in spatial frequency tuning of macaque V1 neurons: effects of a changing receptive field size. *Journal of Neurophysiology*, 88(3), 1363-1373.
- 2- Finn, I. M., Priebe, N. J., & Ferster, D. (2007). The emergence of contrast-invariant orientation tuning in simple cells of cat visual cortex. *Neuron*, 54(1), 137–152.

3) In the methods, the authors state that contrast response functions were assessed per neuron, spanning a nice range of contrasts. Was this done at different luminances as well? The outcome of the contrast (and luminance) increment/decrement manipulation might be expected to vary depending on the contrast sensitivity profile for a given neuron. For instance, if the CRF had high sensitivity, there may be little headroom for detectable changes in response to increments rather than decrements, simply due to the saturating nonlinearity. Same goes with luminance response. If the cortical luminance response function had plateaued given the base luminance tested, one might expect a increment/decrement asymmetry simply due to that nonlinear profile. Perhaps the authors could examine this dependency. It would be equally fascinating if the switch effects were independent of the contrast sensitivity of a neuron.

The reviewer raises some interesting points about the interaction between luminance and contrast and we hope to address some of them in future experiments. We only measured the contrast response function of individual cells with a single mean luminance. Unfortunately, we were limited in our selection of possible stimuli by our desire to combine two luminances and two contrasts. These needed to be chosen to ensure that the majority of neurons on our multielectrode arrays responded to all stimuli. This was a trade-off necessitated by our use of arrays for recording from multiple neurons, along with our aim to collect data associated with all luminance and contrast transitions while maintaining good isolations and responses from as many neurons as possible.

4) Some of the figures could benefit from legends (such as Figure 3), as it was sometimes difficult to decipher what condition a given trace corresponded to.

To make the testing conditions clearer, we have replicated the inset from Figure 1 in Figures 2, 3, 4 and 7, and extended the description of the conditions in the Figure Legends.

Reviewers' comments:

Reviewer #1 (Remarks to the Author):

In my original review, I stated "Despite the strength that comes from a great set of neurophysiological data and strong computational analyses that have merit on their own as reportable, novel scientific findings, the package of results here does not stand well as a coherent set of work supporting the overall thesis of the authors." In the revised manuscript, the authors have added quite a bit of psychophysical data, and indeed that has improved the study. Largely, however, looking at my own and other comments, the authors have not been particularly responsive on a number of points. Specifically, the original reviews raised issues about the modeling, the single neuron vs. population analysis, the suitability of the modeling, and the treatment of other stimulus variables as "nuisance." In addition, although they added new psychophysical data related to pupil changes during luminance and contrast increments and decrements, they did not address the initial concern that the dilating drops in the anesthetized animals invalidates the comparison with human psychophysical data on luminance changes. While I think there have been some nice improvements, my original concerns stand.

Reviewer #2 (Remarks to the Author):

This revised manuscript is more clearly described and is a significant improvement. However, one substantial point raised by reviewer 1 (Pupil size) has not been adequately addressed. It is clear in figure S6 that much of the luminance adaptation seen psychophysically is accomplished by the pupil, even before the end of the "early" window. This is a much bigger problem than the discussion at line 334 points out. There they essentially fall back on the fact that the effects of contrast changes cannot be explained by pupil size. But they don't clarify that the comparison of luminance change effects is deeply problematic.

This may also explain some of the discrepancies that they do not highlight. Because they focus on early vs late comparisons, they do not point out that the psychophysical performance late after a luminance decrement is similar to that later after a luminance increment (Fig 5 D,E) whereas the decoding performance for these two cases is very different (figure 3I). The discussion of figure 5 needs to point out the discrepancies as well as the congruencies. Ideally, they could show a plot like 4A, plotting perceptual accuracy vs population decoding performance for all 16 conditions. This is also related to their answer about commutativity – it's easy to get the impression that things might be non-commutative. This happens because very different responses are simply not compared.

Smaller point. I remain unconvinced that the recurrent network model is valuable. I agree with the authors that the failure of the simple model is informative (but not very surprising – it's hard to see how such a simple model could account for the asymmetries), and so this should be included. But adding the recurrent model really does not help. But I leave it up to them.

Reviewer #3 (Remarks to the Author):

The authors have done an excellent job responding to reviewer comments. This is an interesting study, and I believe warrants publication.

Response to reviewer comments

Three primary concerns and requirements were highlighted by the Editor:

- (1a) adding the single vs population neuron analysis;
- (1b) adding evidence of model suitability;
- (2) addressing the issue with pupil dilation and incorporating Reviewer 2's comments regarding discrepancies potentially explained by that issue; and
- (3) addressing reviewers' issues with our interpretation of treatment variables as "nuisance."

Briefly, we have addressed all of these concerns by:

- (1a) expanding our description of how and why we conduct both single neuron and population analyses in the Results and Discussion sections (Response B);
- (1b) removing all modelling sections (Responses A and G);
- (2) further discussing the implications of our pupillometry results, including possible limitations, but also why these do not invalidate our conclusions (Responses D and E);
- (3) explained the statistical, rather than functional, interpretation of a "nuisance" variable, and removed the word "nuisance" from the title to avoid potential confusion (Response C).

Below, the **reviewer's comments are in bold**, and our responses to each point are in blue. Page numbers refer to the version with Tracked Changes

Collectively, we strongly believe these changes have made the manuscript a coherent and stronger package that combines physiology, computational analyses and psychophysics to shed light on how adaptation to one variable affects encoding of another, a challenge that faces all sensory systems, but which to date has not been studied.

Reviewer #1 (Remarks to the Author):

In my original review, I stated **"Despite the strength that comes from a great set of neurophysiological data and strong computational analyses that have merit on their own as reportable, novel scientific findings, the package of results here does not stand well as a coherent set of work supporting the overall thesis of the authors."** In the revised manuscript, the authors have added quite a bit of psychophysical data, and indeed that has improved the study.

Largely, however, looking at my own and other comments, the authors have not been particularly responsive on a number of points. Specifically, the original reviews raised issues about the modeling (see Response A), the single neuron vs. population analysis (see Response B), the suitability of the modeling, and the treatment of other stimulus variables as "nuisance" (see Response C).

Response A

Reflecting on the comments from Reviewers 1 & 2, we agree that the modeling makes a minor contribution to the overall manuscript and dilutes the impact of the package. Therefore, we have removed all modelling sections from the manuscript, including text in the abstract, introduction, results (notably, pages 20-22 and Figure 7), discussion and methods. Reflecting these substantial changes, we have also added a new final

paragraph to the manuscript (page 27), which has been copied below. We believe that the physiological and psychophysical experiments stand well as a coherent set of findings that support our overall thesis.

“Collectively, we have shown that orientation discriminability is luminance and contrast dependent, changing over time due to firing rate adaptation. This is accompanied by changes in the information available about orientation from neural spiking, attributable to adaptive gain modulation. Multiple cellular and network mechanisms may account for adaptive gain control in the cortex. At the level of single neurons, adaptation largely relies on cellular mechanisms (e.g., Na⁺-activated and Ca²⁺-activated K⁺ currents)⁵⁰ and synaptic depression⁵¹. Adaptive mechanisms can also be derived from network dynamics and recurrent inhibition, which can produce neuronal response dynamics that vary over a range of time scales^{3,52}. In a highly interconnected network such as V1, lateral inhibition or modulatory feedback can also account for adaptive gain control^{33,34,53,54}. Future studies should identify the distinct contributions of inhibitory and excitatory neurons to the changing feature selectivity that occurs during adaptation, in order to link single-neuron and circuit-level mechanisms to perceptual outcomes.”

Response B

Unfortunately, we are not certain which previous comment the reviewer refers to in highlighting an issue with “the single neuron vs. population analysis”. Based on our understanding, we assume that the reviewer refers to a comment raised by Reviewer #2 in the first round of reviews:

Does the population decoding reveal anything that is not seen at the single neuron level? It seems like both analyses give the same result. Which is fine, but it’s not made clear for the reader that these are just two descriptions of the same phenomenon.

In sum, the population decoding does not reveal anything not seen at the single neuron level, however, this was not a foregone conclusion given the dynamic sensitivity of individual neurons. This population analysis constitutes a single figure panel in the paper (Fig. 3I), and we believe is an intuitive way to summarize the global effects of adaptation on orientation coding and provides a simple metric that can be compared with human perceptual performance.

Single neurons in our population are heterogeneous; they have different tuning preferences, adaptive properties, and temporal dynamics. The aim of the population decoding was to investigate whether we can track changes in the population code for stimulus orientation during adaptation, despite the heterogeneity at the single neuron level. To address the reviewers’ concerns, we now describe the rationale for performing the population-level analyses (pages 12-13):

“We found that the information conveyed by that individual neurons about stimulus orientation is strongly affected during adaptation. It is, however, unclear how the coding of orientation by a neural population is affected by changes over time in firing rates and trial-to-trial variability at the level of individual neurons, and whether the variability between neurons can be overcome at the population level. Moreover, neurons vary in their preferences, temporal dynamics, and adaptation properties. Therefore, we used a population decoding approach to quantify orientation discriminability, asking how decoding accuracy: (1) is affected by different luminance and contrast conditions; and (2) how it changes during the course of adaptation to a single luminance and contrast.”

Subsequently, we highlight that the single-neuron and population-level approaches produce similar results by adding the following to the end of the associated Results paragraph (page 13):

“Reflecting the single neuron analyses, this demonstrates that coding of stimulus orientation by neural populations is substantially affected by adaptation to luminance and contrast, and this adaptive coding tracks the direction of changes in spiking rate, not response variability.”

Finally, we have added a sentence to the Discussion (page 27):

“A simple way to address this issue is using population decoding to investigate if the observed effects of adaptation at the single neuron level are also evident at the population, or circuit, level. Our analysis showed that the decoding accuracy is significantly luminance- and contrast-dependent and strongly affected by adaptation.”

Response C

In their initial reviews, Reviewer 1 stated:

“An overarching issue that’s not addressed is that luminance and contrast are only nuisances from the perspective of a decoder that wants to know about orientation, but from the perspective of the animal they are alternative stimulus variables. The system does actually have information about the true luminance and contrast on some timescales (i.e., it would be decodable). The authors have a clear goal for their study which focuses on orientation, which is fine and appropriate given what we know about luminance/contrast coding in the earliest stages of visual processing, but this issue should at least be discussed.”

Further, Reviewer 3 stated:

Perhaps this is just semantics, but the reference to contrast and luminance as “nuisance variables” both in the title and main manuscript seems odd. Particularly since the point of the study is to reveal that they are anything but nuisance variables. But also because they are usually considered stimulus parameters of main interest in many other studies. The authors might want to reconsider that framing.

We use the term “nuisance variable” according to its formal meaning within the fields of inferential statistics and decoding. There, it refers to variables (e.g. contrast) that impact on a set of measurements (e.g. neural activity) but which can ideally be factored out or ignored because they are not relevant to a particular inference (e.g. inferring orientation by decoding the neural activity). “Nuisance” is not intended to imply that these variables are unimportant to the brain or behaviour, however, we agree that there is room for confusion with the colloquial definition of “nuisance”, particularly in the title where there is little context. Accordingly, we have made numerous changes to the way we use and define the word “nuisance”.

First, we have removed the word nuisance from the title and re-named the paper to “Contrast and luminance adaptation alter neuronal and perceptual coding of stimulus orientation”.

In the Introduction, we now describe our treatment of contrast and luminance with respect to our decoding paradigm (page 3):

“This experimental design allowed us to investigate how adaptation to one dimension (luminance or contrast) affects neural coding of another dimension (orientation). Our study is the first to reveal how orientation coding in V1 neurons is impacted by adaptation to presumably orthogonal stimulus dimensions. Although the encoding of luminance and contrast are critical functions of the visual system, here we are interested specifically in the encoding of orientation during adaptation; therefore we treat luminance and contrast as nuisance variables in the statistical sense.”

In the Discussion, we also discuss how this perspective on luminance and contrast may be carried forward to future work, conducted from a different perspective (page 25, Discussion):

“The aim of our study was to explore the effect of adaptation to luminance and contrast on the coding of stimulus orientation. From the perspective of encoding stimulus orientation, changes in luminance and contrast are problematic, making them “nuisance variables”. Despite this, neurons in most visual areas actually carry information about luminance and contrast, and this information can be reliably decoded. While not the focus of our study, this is evident in our data as differences in the steady-state firing rate after adaptation to luminance and contrast (e.g. the mean firing rates for low and high contrasts are different and thus decodable).”

Response D

In addition, although they added new psychophysical data related to pupil changes during luminance and contrast increments and decrements, they did not address the initial concern that the dilating drops in the anesthetized animals invalidates the comparison with human psychophysical data on luminance changes. While I think there have been some nice improvements, my original concerns stand.

We appreciate the reviewer following up on this and acknowledge that we misinterpreted the intent of their initial comments. We now specifically describe the limitations of our psychophysical studies, highlight the fact that pupil changes can occur in the humans but not the animals, and discuss how these psychophysical data should best be considered. Critically, our overall position remains as before: pupil size modulations cannot explain the adaptive neural and perceptual phenomena that accompany contrast changes, and are unlikely to account for the adaptive phenomena following luminance changes because of the different timescales of the effects.

In the Results, following the description of the human psychophysical results, we now state (page 15):

“These results are broadly consistent with our physiological data, where neural population decoding accuracy was significantly higher during the early versus the late phase following a contrast increment and luminance decrement (**Figure 3**). Despite this, an important methodological difference between the physiological and psychophysical studies limits straight-forward comparison of their results. In our physiological study, we used phenylephrine and atropine eye drops to dilate the pupils. This inactivates the pupillary light reflex, which modulates the amount of light reaching the retina.

We then describe the results of the pupillometry and have added the following caveat (page 15-16):

“Contrast increments are therefore associated with consistent changes in perceptual and neuronal discrimination performance, in the absence of associated pupillary changes. However, while luminance decrements were also associated with consistent changes in perceptual and neuronal discrimination, they were only accompanied by pupillary dilation in the human observers. This means that although the observed changes in neural coding can only arise as a result of cascading neural processes in the visual hierarchy (because the pupils were permanently dilated in the marmosets), the changes in human perceptual performance could reflect these same neural processes, or simply the effects of pupillary dilation.”

Finally, in the Discussion, we have added (page 27):

“While we cannot rule out a role for pupil size variations in the perceptual luminance adaptation effects we observed, two factors make it likely that the neural adaptation observed following both contrast increments and luminance decrements can account for the changes in human

psychophysical performance. First, we observed a significant difference in perceptual thresholds between the early and late phases following contrast increments, even though the pupil size remained mostly unchanged. Second, following a luminance increment, pupil size changed substantially over time, but we observed relatively little change in discrimination thresholds.”

Reviewer #2 (Remarks to the Author):

This revised manuscript is more clearly described and is a significant improvement. However, one substantial point raised by reviewer 1 (Pupil size) has not been adequately addressed. It is clear in figure S6 that much of the luminance adaptation seen psychophysically is accomplished by the pupil, even before the end of the “early” window. This is a much bigger problem than the discussion at line 334 points out. There they essentially fall back on the fact that the effects of contrast changes cannot be explained by pupil size. But they don’t clarify that the comparison of luminance change effects is deeply problematic.

Response E

These concerns are addressed in detail in Response D to Reviewer #1.

In brief, we have substantially changed the description of our psychophysical results, and how we describe our ability to link the human psychophysical and monkey physiological results in the case of the luminance changes.

This may also explain some of the discrepancies that they do not highlight. Because they focus on early vs late comparisons, they do not point out that the psychophysical performance late after a luminance decrement is similar to that later after a luminance increment (Fig 5 D,E) whereas the decoding performance for these two cases is very different (figure 3I). The discussion of figure 5 needs to point out the discrepancies as well as the congruencies. Ideally, they could show a plot like 4A, plotting perceptual accuracy vs population decoding performance for all 16 conditions. This is also related to their answer about commutativity – its easy to get the impression that things might be non-commutative. This happens because very different responses are simply not compared.

Response F

The key finding of our study is that the coding of stimulus orientation (at the neural and perceptual level) changes during adaptation to a nuisance variable (i.e. following a luminance or contrast switch). In addition, the results of our physiological and psychophysical experiments are suggestive of commutativity between the late responses to different types of switch (see below). However, our main point in this paper was not to make this comparison, which was the focus of an earlier manuscript, in which our experiments were explicitly designed to facilitate this comparison (Ghodrati, Alwis & Price, 2016).

We appreciated the suggestion of the new analysis comparing human perceptual discrimination thresholds against marmoset neuronal population decoding performance and have copied it below. Note that there are only 8 conditions (2 time-points following 4 possible switches tested psychophysically). A simple prediction here is that we should expect a negative correlation across the 8 data points – lower perceptual thresholds correspond to higher neuronal decoding accuracy. This is clearly not evident when all 8 points are considered. If only Early versus Late conditions are considered following a change (connected pairs of points), the expected negative correlation is seen following contrast increments (solid blue) and luminance decrements (dashed brown). This negative trend is not evident following luminance increments and contrast decrements, but the differences in both accuracy and threshold are much smaller here (note also the high variability of the data points for the psychophysical thresholds).

The reviewer suggests additional comparisons, between the same time points but following two different stimulus changes. The graph shows that on average, subjects had lower discrimination thresholds (better performance) in the late phase following a luminance decrement (mean=1.7) than the late phase following a luminance increment (mean=2.2); see horizontal brown arrows. Although this difference is not significant, it is consistent with the physiology (vertical arrows).

Note that testing of the effects of different luminance and contrast changes were conducted with different participants, and in different experimental blocks or sessions. Therefore, we are extremely cautious about directly comparing performance between any conditions other than Early versus Late for the same switch condition. This makes it difficult to say anything substantive about commutativity (similarities or differences between discrimination performance late after two types of switch).

In summary, while we agree that the differences in perceptual thresholds in different conditions might not be as strong as those of population decoding accuracy, and we still believe that the results of the psychophysical and physiological experiments are broadly consistent with each other, fine-grained quantitative comparisons of the neural and perceptual data are not appropriate here.

Smaller point. I remain unconvinced that the recurrent network model is valuable. I agree with the authors that the failure of the simple model informative (but not very surprising – its hard to see how such a simple model could account for the asymmetries), and so this should be included. But adding the recurrent model really does not help. But I leave it up to them

Response G

As both reviewers raised doubts about whether the modelling was necessary, we have removed it from the manuscript. The associated changes are described in more detail in the first response to Reviewer #1.

Reviewers' comments:

Reviewer #1 (Remarks to the Author):

In this new revision, the authors have removed the modeling results, and made some other improvements/clarifications. In my initial review, I stated that the package of results did not stand well as a coherent set of work. The main strength of this work is the neuronal data, and the analyses there are solid. The modeling was weakly connected, and has been removed. The psychophysical data do not match well to a study of contrast and luminance adaptation, because the pupil is dilated in the animal subjects. I'm not sure how to get past that fact. Although the authors now acknowledge this issue and list it as a caveat, it seems a big stretch to compare dilated, anesthetized animals to humans with an intact pupillary light reflex in a study of luminance adaptation.

Reviewer #2 (Remarks to the Author):

This new revision is strengthened in several ways. The discussion of the psychophysics and pupil size is now much clearer, and I think that removing the recurrent model has also helped. They also now say a little more about the relationship between the results at the level of single neurons and of the population decoder. I still think this final comparison would be stronger if they made it quantitative (I suppose I should have been clearer about this). Currently, the single neuron data are plotted as changes in mutual information (in absolute terms), while the population decoding is in terms of % performance. These are not commensurate measures, so if the effect at the population level were 10x larger than one neuron at a time, we would not know. Does "reflecting the single neuron analysis" mean that this ratio is $< 2? < 1.1?$ It seems to me that this is a relatively simple thing to do add – for each if the calculated the ratio of mutual information (early vs late) instead of the difference, then a single summary ratio for the single unit data could be compared with the population. (This would not require new figures – the current histogram is fine. But calculating a summary ratio would help). This is a small point, and at this point I am happy to leave this to the authors' judgement, although obviously I encourage them to include this extra calculation.

Contrast and luminance adaptation alter neuronal coding and perception of stimulus orientation

Masoud Ghodrati, Elizabeth Zavitz, Marcello GP Rosa, Nicholas SC Price

We believe that our addition of new behavioural experiments and analysis in the previous revision, accompanied by the caveats in the Results and Discussion regarding the effects of pupil size changes in the psychophysical data, have already addressed Reviewer 1's comments about "the comparison between animals and humans and the effect of pupil changes".

We have added "the quantitative version of the single vs population neuron analysis" suggested by Reviewer 2.

Below, we copy **Reviewer 2's comments**, followed by our response in blue. Page numbers refer to the version with Tracked Changes

Reviewer #2 (Remarks to the Author):

This new revision is strengthened in several ways. The discussion of the psychophysics and pupil size is now much clearer, and I think that removing the recurrent model has also helped. They also now say a little more about the relationship between the results at the level of single neurons and of the population decoder. I still think this final comparison would be stronger if they made it quantitative (I suppose I should have been clearer about this). Currently, the single neuron data are plotted as changes in mutual information (in absolute terms), while the population decoding is in terms of % performance. These are not commensurate measures, so if the effect at the population level were 10x larger than one neuron at a time, we would not know. Does "reflecting the single neuron analysis" mean that this ratio is $< 2? < 1.1?$ It seems to me that this is a relatively simple thing to do add – for each if the calculated the ratio of mutual information (early vs late) instead of the difference, then a single summary ratio for the single unit data could be compared with the population. (This would not require new figures – the current histogram is fine. But calculating a summary ratio would help).

This is a small point, and at this point I am happy to leave this to the authors' judgement, although obviously I encourage them to include this extra calculation.

Response

The Reviewer's point is well received, as it provides a useful way to highlight the similarity of the effects of adaptation at the single-neuron and population level. On page 13, we have added the following paragraph:

"To compare the size of the changes in orientation coding at the single neuron and population levels, we calculated a mutual information ratio (early relative to late) for each stimulus switch, and compared it to the corresponding ratio of decoding performance (early relative to late). Across the 8 switches, these ratios were strongly correlated ($r=0.87$, $p=0.004$), with a slope of 0.4. Inasmuch as these ratios can be compared, this suggests that the adaptive changes over time are of similar magnitude at the single neuron and population levels. Further, coding of stimulus orientation by neural populations is substantially affected by adaptation to luminance and contrast, and this adaptive coding tracks the direction of changes in spiking rate, not response variability."

Reviewer #2 (Remarks to the Author):

I am happy to recommend publication now.